# Nonparametric Classification on Low Dimensional Manifolds using Overparameterized Convolutional Residual Networks

**Zixuan Zhang**[*]
Georgia Tech
zzhang3105@gatech.edu

**Kaiqi Zhang**[*]
UC Santa Barbara
kzhang70@ucsb.edu

**Minshuo Chen**
Northwestern University
minshuo.chen@northwestern.edu

**Yuma Takeda**
University of Tokyo
utklav1511@gmail.com

**Mengdi Wang**
Princeton University
mengdiw@princeton.edu

**Tuo Zhao**
Georgia Tech
tourzhao@gatech.edu

**Yu-Xiang Wang**
UC San Diego
yuxiangw@ucsd.edu

## Abstract

Convolutional residual neural networks (ConvResNets), though *overparameterized*, can achieve remarkable prediction performance in practice, which cannot be well explained by conventional wisdom. To bridge this gap, we study the performance of ConvResNeXts trained with weight decay, which cover ConvResNets as a special case, from the perspective of nonparametric classification. Our analysis allows for infinitely many building blocks in ConvResNeXts, and shows that weight decay implicitly enforces sparsity on these blocks. Specifically, we consider a smooth target function supported on a low-dimensional manifold, then prove that ConvResNeXts can adapt to the function smoothness and low-dimensional structures and efficiently learn the function without suffering from the curse of dimensionality. Our findings partially justify the advantage of *overparameterized* ConvResNeXts over conventional machine learning models.

## 1 Introduction

Deep learning has achieved significant success in various real-world applications, such as computer vision [14, 23, 26], natural language processing [2, 15, 42], and robotics [16]. One notable example of this is in the field of image classification, where the winner of the 2017 ImageNet challenge achieved a top-5 error rate of just 2.25% [19] using Convolutational Residual Network (ConvResNets) on a training dataset of 1 million labeled high-resolution images in 1000 categories.

Researchers have attributed the remarkable performance of deep learning to its great flexibility in modeling complex functions, which has motivated many works on investigating the representation power of deep neural networks. For instance, early work such as Barron [3], Cybenko [7], Kohler and Krzyżak [22] initialized this line of research for simple feedforward neural networks (FNNs) [19, 20, 38, 37]. More recently, Suzuki [32], Yarotsky [41] gave more precise bounds on the model sizes in terms of the approximation error, and Oono and Suzuki [30] further established a bound for more advanced architectures – ConvResNets. Based on these function approximation theories,

---

[*]Equal contribution.

38th Conference on Neural Information Processing Systems (NeurIPS 2024).

one can further establish generalization bounds of deep neural networks with finite samples. Taking Oono and Suzuki [30] as an example again, they showed that ConvResNets with $\tilde{O}(n^{D/(2\alpha+D)})$ parameters can achieve a minimax optimal convergence rate $\tilde{O}(n^{-2\alpha/(2\alpha+D)})$ while approximating a $C^\alpha$ nonparametric regression function with $n$ samples. Unfortunately, these theoretical results cannot well explain the empirical successes of deep learning well, as they require the model size to be no larger than $\tilde{O}(n)$ (the generalization bounds become vacuous otherwise). However, in real applications, practical deep learning models are often overparmameterized, that is the model size can greatly exceeds the sample size.

## 1.1 Main Results

Overparameterization of neural networks has been considered as one of the most fundamental research problems in deep learning theories, where parameters can significantly exceed training samples. There has been substantial empirical evidence showing that overparameterization can help fit the training data, ease the challenging nonconvex optimization, and gain robustness. However, existing literature on deep learning theories under such an **overparameterized** regime is very limited. To the best of our knowledge, we are only aware of Zhang and Wang [43], which attempts to analyze overparameterized neural networks trained with weight decay. However, their work still suffers from two major restrictions: (1) They consider parallel FNN, which is rarely used in practice. Whether similar results hold for more practical architectures remains unclear; (2) Their generalization bound from the curse of dimensionality, where the sample size is require to scale exponentially with the input dimension.

To address (1), we propose to develop a new theory for nonparametric classification using overparameterized ConvResNeXts trained with weight decay [40]. The ConvResNeXt generalizes ConvResNets and includes them as a special case [5, 18, 33, 44]. Compared with FNNs, ConvResNeXts exhibit three features: (i) Instead of using dense weight matrices, they use convolutional filters, which can naturally investigate the underlying structures of the input data such as images and acoustic signals; (ii) They are equipped with skip-layer connections, which divides the entire network into blocks. The skip-layer connection can effectively address the vanishing gradient issue and therefore allow the networks to be significantly deeper; (iii) They are equipped with parallel architectures, which enable multiple "paths" within each block of the network, and allows the network to learn a more diverse set of features. Figure 1b illustrates the structure of ConvResNeXts (detailed introductions of ConvResNeXts is deferred to Section 2.3). This architecture introduces a significantly more complex nested function form, presenting us with the challenge of addressing novel issues in bounding the metric entropy of the function class.

To address (2), our proposed theory considers the optimal classifier is supported on a $d$-dimensional smooth manifold $\mathcal{M}$ isometrically embedded in $\mathbb{R}^D$ with $d \ll D$. The low-dimensional manifold assumption is highly practical, since it aligns with the inherent nature of many real-world datasets. For example, images typically represent projections of 3-dimensional objects subject to various transformations like rotation, translation, and skeletal adjustments. Such a generating mechanism inherently involves a limited set of intrinsic parameters. More broadly, various forms of data, including visual, acoustic, and textual, often exhibit low dimensional structures due to rich local regularities, global symmetries, repetitive patterns, or redundant sampling. It is reasonable to model these data as samples residing in proximity to a low dimensional manifold.

Our theoretical results can be summarized as follows:

- We prove that when ConvResNeXts are overparameterized, i.e., the number of blocks is larger than the order of the sample size $n$, they can still achieve an asymptotic minimax rate for learning Besov functions when trained with weight decay. That is, given that the target function belongs to the Besov space $B_{p,q}^\alpha(\mathcal{M})^2$, the risk of the estimator given by the ConvResNeXt class converges to the optimal risk at the rate $\tilde{O}(n^{-\frac{\alpha/d}{2\alpha/d+1}(1-o(1))})$ with $n$ samples. Notably, the statistical rate of convergence in our theory only depends on the intrinsic dimension $d$, which circumvents the curse of dimensionality in Zhang and Wang [43].

---

[2] The Besov space includes functions with spatially heterogeneous smoothness and generalizes more elementary function spaces such as Sobolev and Hölder spaces.

• Moreover, our theory shows that one can scale the number of "paths" $M$ in each block with the depth $N$ as roughly $MN \gtrsim n^{\frac{1}{2\alpha/d+1}}$, which does not affect the convergence rate. This partially justifies the flexibility of the ConvResNeXt architecture when designing the bottlenecks, which simple structures like FNNs **cannot** achieve. Moreover, we can exchange the number of "paths" $M$ and depth $N$ as long as their product remains the same. This further provides the architectural insight that we don't necessarily need parallel blocks when we have residual connections. To say it differently, we provide new insight into why "residual connection" and "parallel blocks" in ResNeXts are useful in both approximation and generalization.

• Another technical highlight of our paper is bounding the covering number of weight-decayed ConvResNeXts, which is essential for computing the critical radius of the local Gaussian complexity. Specifically, we adopted a more advanced method that leverages the Dudley's chaining of the metric entropy [4]. This technique provides a tighter bound than choosing a single radius of the covering number as in Zhang and Wang [43].

• To the best of our knowledge, our work is the first to develop approximation and statistical theories for ConvResNeXts, as well as overparameterized ConvResNets.

## 1.2 Related Works

Our work is closely related to Liu et al. [25], which studies nonparametric classification under a similar setup – the optimal classifier belongs to the Besov space supported on a low dimensional manifold. Despite they develop similar theoretical results to ours, their analysis does not allow the model to be overparameterized, and therefore is not applicable to practical neural networks. Moreover, they investigate ConvResNets, which is a special case of ConvResNeXt in our work.

Our work is closely related to the reproducing kernel methods, which are also often used for nonparametric regression. However, existing literature has shown that the reproducing kernel methods lack the adaptivity to handle the heterogeneous smoothness in estimating Besov space functions, and only achieve suboptimal rate of convergence in statistical estimation [9, 32].

Our work is closely related neural tangent kernel theories [21, 1], which study overparameterized neural networks. Specifically, under certain regularity conditions, they establish the equivalence between overparameterized neural networks and reproducing kernel methods, and therefore the generalization bounds of overparameterized networks can be derived based on the associated reproducing kernel Hilbert space. Note that neural tangent kernel theories can be viewed as special cases of the theories for general reproducing kernel methods. Therefore, they also lack the adaptivity to be successful in the Besov space thus do not capture the properties of overparameterized neural networks.

## 2 Preliminaries

In this section, we introduce some concepts on manifolds. Details can be found in [35] and [24]. Then we provide a detailed definition of the Besov space on smooth manifolds and the ConvResNeXt architecture.

### 2.1 Smooth manifold

Firstly, we briefly introduce manifolds, the partition of unity and reach. Let $\mathcal{M}$ be a $d$-dimensional Riemannian manifold isometrically embedded in $\mathbb{R}^D$ with $d$ much smaller than $D$.

**Definition 2.1** (Chart). A chart on $\mathcal{M}$ is a pair $(U, \phi)$ such that $U \subset \mathcal{M}$ is open and $\phi : U \mapsto \mathbb{R}^d$, where $\phi$ is a homeomorphism (i.e., bijective, $\phi$ and $\phi^{-1}$ are both continuous).

In a chart $(U, \phi)$, $U$ is called a coordinate neighborhood, and $\phi$ is a coordinate system on $U$. Essentially, a chart is a local coordinate system on $\mathcal{M}$. A collection of charts that covers $\mathcal{M}$ is called an atlas of $\mathcal{M}$.

**Definition 2.2** ($C^k$ Atlas). A $C^k$ atlas for $\mathcal{M}$ is a collection of charts $\{(U_i, \phi_i)\}_{i \in \mathcal{A}}$ which satisfies $\bigcup_{i \in \mathcal{A}} U_i = \mathcal{M}$, and are pairwise $C^k$ compatible:

$$\phi_i \circ \phi_\beta^{-1} : \phi_\beta(U_i \cap U_\beta) \rightarrow \phi_i(U_i \cap U_\beta) \text{ and } \phi_\beta \circ \phi_i^{-1} : \phi_i(U_i \cap U_\beta) \rightarrow \phi_\beta(U_i \cap U_\beta)$$

are both $C^k$ for any $i, \beta \in \mathcal{A}$. An atlas is called finite if it contains finitely many charts.

**Definition 2.3** (Smooth Manifold). A smooth manifold is a manifold $\mathcal{M}$ together with a $C^\infty$ atlas.

Classical examples of smooth manifolds are the Euclidean space, the torus, and the unit sphere. Furthermore, we define $C^s$ functions on a smooth manifold $\mathcal{M}$ as follows:

**Definition 2.4** ($C^s$ functions on $\mathcal{M}$). Let $\mathcal{M}$ be a smooth manifold and $f : \mathcal{M} \to \mathbb{R}$ be a function on $\mathcal{M}$. A function $f : \mathcal{M} \to \mathbb{R}$ is $C^s$ if for any chart $(U, \phi)$ on $\mathcal{M}$, the composition $f \circ \phi^{-1} : \phi(U) \to \mathbb{R}$ is a continuously differentiable up to order $s$.

We next define the $C^\infty$ partition of unity, which is an important tool for studying functions on manifolds.

**Definition 2.5** (Partition of Unity, Definition 13.4 in [35]). A $C^\infty$ partition of unity on a manifold $\mathcal{M}$ is a collection of $C^\infty$ functions $\{\rho_i\}_{i \in \mathcal{A}}$ with $\rho_i : \mathcal{M} \to [0, 1]$ such that for any $\boldsymbol{x} \in \mathcal{M}$, there is a neighbourhood of $\boldsymbol{x}$ where only a finite number of the functions in $\{\rho_i\}_{i \in \mathcal{A}}$ are nonzero, and

$$\sum_{i \in \mathcal{A}} \rho_i(\boldsymbol{x}) = 1.$$

An open cover of a manifold $\mathcal{M}$ is called locally finite if every $\boldsymbol{x} \in \mathcal{M}$ has a neighborhood that intersects with a finite number of sets in the cover. The following proposition shows that a $C^\infty$ partition of unity for a smooth manifold always exists.

**Proposition 2.6** (Existence of a $C^\infty$ partition of unity, Theorem 13.7 in [35]). *Let $\{U_i\}_{i \in \mathcal{A}}$ be a locally finite cover of a smooth manifold $\mathcal{M}$. Then there is a $C^\infty$ partition of unity $\{\rho_i\}_{i=1}^\infty$ where every $\rho_i$ has a compact support such that $\mathrm{supp}(\rho_i) \subset U_i$.*

Let $\{(U_i, \phi_i)\}_{i \in \mathcal{A}}$ be a $C^\infty$ atlas of $\mathcal{M}$. Proposition 2.6 guarantees the existence of a partition of unity $\{\rho_i\}_{i \in \mathcal{A}}$ such that $\rho_i$ is supported on $U_i$. To characterize the curvature of a manifold, we adopt the geometric concept: reach.

**Definition 2.7** (Reach [12, 29]). Denote

$$G = \left\{ \boldsymbol{x} \in \mathbb{R}^D : \exists\, \boldsymbol{p} \neq \boldsymbol{q} \in \mathcal{M} \text{ such that } \|\boldsymbol{x} - \boldsymbol{p}\|_2 = \|\boldsymbol{x} - \boldsymbol{q}\|_2 = \inf_{\boldsymbol{y} \in \mathcal{M}} \|\boldsymbol{x} - \boldsymbol{y}\|_2 \right\}$$

as the set of points with at least two nearest neighbors on $\mathcal{M}$. The closure of $G$ is called the medial axis of $\mathcal{M}$. Then the reach of $\mathcal{M}$ is defined as

$$\tau = \inf_{\boldsymbol{x} \in \mathcal{M}} \inf_{\boldsymbol{y} \in G} \|\boldsymbol{x} - \boldsymbol{y}\|_2.$$

Reach has a simple geometrical interpretation: for every point $\boldsymbol{x} \in \mathcal{M}$, the osculating circle's radius is at least $\tau$. A large reach for $\mathcal{M}$ indicates that the manifold changes slowly.

## 2.2 Besov functions on a smooth manifold

We next define the Besov function space on the smooth manifold $\mathcal{M}$, which generalizes more elementary function spaces such as the Sobolev and Hölder spaces. Roughly speaking, functions in the Besov space are only required to have weak derivatives with bounded total variation. Notably, this includes functions with spatially heterogeneous smoothness, which requires more locally adaptive methods to achieve optimal estimation errors [10]. Examples of Besov class functions include piecewise linear functions and piecewise quadratic functions that are smoother in some regions and more wiggly in other regions; see e.g., Figure 2 and Figure 4 of Mammen and van de Geer [27].

To define Besov functions rigorously, we first introduce the modulus of smoothness.

**Definition 2.8** (Modulus of Smoothness [8, 32]). Let $\Omega \subset \mathbb{R}^D$. For a function $f : \mathbb{R}^D \to \mathbb{R}$ be in $L^p(\Omega)$ for $p > 0$, the $r$-th modulus of smoothness of $f$ is defined by

$$w_{r,p}(f, t) = \sup_{\|\boldsymbol{h}\|_2 \leq t} \|\Delta_{\boldsymbol{h}}^r(f)\|_{L^p},$$

$$\text{where } \Delta_{\boldsymbol{h}}^r(f)(\boldsymbol{x}) = \begin{cases} \sum_{j=0}^r \binom{r}{j} (-1)^{r-j} f(\boldsymbol{x} + j\boldsymbol{h}) & \text{if } \boldsymbol{x}, \boldsymbol{x} + r\boldsymbol{h} \in \Omega, \\ 0 & \text{otherwise.} \end{cases}$$

**Definition 2.9** (Besov Space $B_{p,q}^{\alpha}(\Omega)$). For $0 < p, q \leq \infty, \alpha > 0, r = \lfloor \alpha \rfloor + 1$, define the seminorm $|\cdot|_{B_{p,q}^{\alpha}}$ as

$$|f|_{B_{p,q}^{\alpha}(\Omega)} := \begin{cases} \left( \int_0^{\infty} (t^{-\alpha} w_{r,p}(f,t))^q \dfrac{dt}{t} \right)^{\frac{1}{q}} & \text{if } q < \infty, \\ \sup_{t>0} t^{-\alpha} w_{r,p}(f,t) & \text{if } q = \infty. \end{cases}$$

The norm of the Besov space $B_{p,q}^s(\Omega)$ is defined as $\|f\|_{B_{p,q}^{\alpha}(\Omega)} := \|f\|_{L^p(\Omega)} + |f|_{B_{p,q}^{\alpha}(\Omega)}$. Then the Besov space is defined as $B_{p,q}^{\alpha}(\Omega) = \{f \in L^p(\Omega) | \|f\|_{B_{p,q}^{\alpha}} < \infty\}$.

Moreover, we show that functions in the Besov space can be decomposed using B-spline basis functions in the following proposition.

**Proposition 2.10** (Decomposition of Besov functions). *Any function $f$ in the Besov space $B_{p,q}^{\alpha}, \alpha > d/p$ can be decomposed using B-spline of order $m, m > \alpha$: for any $\boldsymbol{x} \in \mathbb{R}^d$, we have*

$$f(\boldsymbol{x}) = \sum_{k=0}^{\infty} \sum_{\boldsymbol{s} \in J(k)} c_{k,\boldsymbol{s}}(f) M_{m,k,\boldsymbol{s}}(\boldsymbol{x}), \tag{1}$$

*where $J(k) := \{2^{-k}\boldsymbol{s} : \boldsymbol{s} \in [-m, 2^k + m]^d \subset \mathbb{Z}^d\}$, $M_{m,k,\boldsymbol{s}}(\boldsymbol{x}) := M_m(2^k(\boldsymbol{x} - \boldsymbol{s}))$, and $M_k(\boldsymbol{x}) = \prod_{i=1}^d M_k(x_i)$ is the cardinal B-spline basis function which can be expressed as a polynomial:*

$$M_m(z) = \frac{1}{m!} \sum_{j=1}^{m+1} (-1)^j \binom{m+1}{j} (z-j)_+^m. \tag{2}$$

We next define $B_{p,q}^{\alpha}$ functions on $\mathcal{M}$.

**Definition 2.11** ($B_{p,q}^{\alpha}$ Functions on $\mathcal{M}$ [13, 34]). Let $\mathcal{M}$ be a compact smooth manifold of dimension $d$. Let $\{(U_i, \phi_i)\}_{i=1}^{C_{\mathcal{M}}}$ be a finite atlas on $\mathcal{M}$ and $\{\rho_i\}_{i=1}^{C_{\mathcal{M}}}$ be a partition of unity on $\mathcal{M}$ such that $\text{supp}(\rho_i) \subset U_i$. A function $f : \mathcal{M} \to \mathbb{R}$ is in $B_{p,q}^{\alpha}(\mathcal{M})$ if

$$\|f\|_{B_{p,q}^{\alpha}(\mathcal{M})} := \sum_{i=1}^{C_{\mathcal{M}}} \|(f\rho_i) \circ \phi_i^{-1}\|_{B_{p,q}^{\alpha}(\mathbb{R}^d)} < \infty. \tag{3}$$

Since $\rho_i$ is supported on $U_i$, the function $(f\rho_i) \circ \phi_i^{-1}$ is supported on $\phi(U_i)$. We can extend $(f\rho_i) \circ \phi_i^{-1}$ from $\phi(U_i)$ to $\mathbb{R}^d$ by setting the function to be 0 on $\mathbb{R}^d \setminus \phi(U_i)$. The extended function lies in the Besov space $B_{p,q}^s(\mathbb{R}^d)$ [34, Chapter 7].

## 2.3 Architecture of ConvResNeXt

We introduce the architecture of ConvResNeXts. ConvResNeXts have three main features: convolution kernel, residual connections, and parallel architecture.

Consider one-sided stride-one convolution in our network. Let $\mathcal{W} = \{\mathcal{W}_{j,k,l}\} \in \mathbb{R}^{w' \times K \times w}$ be a convolution kernel with output channel size $w'$, kernel size $K$ and input channel size $w$. For $\boldsymbol{z} \in \mathbb{R}^{D \times w}$, the convolution of $\mathcal{W}$ with $\boldsymbol{z}$ gives $\boldsymbol{y} \in \mathbb{R}^{D \times w'}$ such that

$$\boldsymbol{y} = \mathcal{W} \star \boldsymbol{z}, \quad y_{i,j} = \sum_{k=1}^K \sum_{l=1}^w \mathcal{W}_{j,k,l} z_{i+k-1,l}, \tag{4}$$

where $1 \leq i \leq D, 1 \leq j \leq w'$ and we set $z_{i+k-1,l} = 0$ for $i + k - 1 > D$, as demonstrated in Figure 1a.

The building blocks of ConvResNeXts are residual blocks. Given an input $\boldsymbol{x}$, each residual block computes $\boldsymbol{x} + F(\boldsymbol{x})$, where $F$ is a subnetwork called bottleneck, consisting of convolutional layers. In ConvResNeXts, a parallel architecture is introduced to each building block, which enables multiple "paths" in each block. In this paper, we study the ConvResNeXts with rectified linear unit (ReLU) activation function, i.e., $\sigma(z) = \max\{z, 0\}$. We next provide the detailed definition of ConvResNeXts as follows:

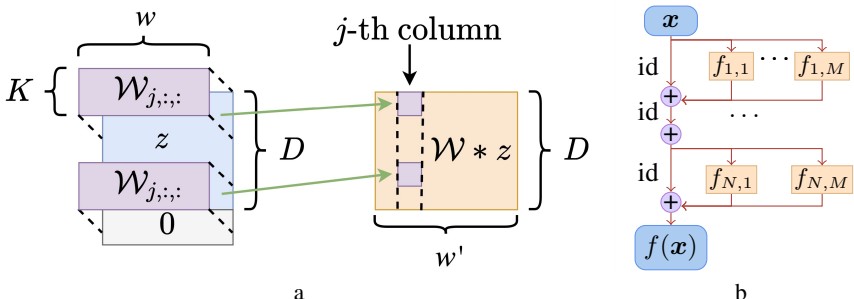

Figure 1: (a) Demonstration of the convolution operation $\mathcal{W} * z$, where the input is $z \in \mathbb{R}^{D \times w}$, and the output is $\mathcal{W} * z \in \mathbb{R}^{D \times w'}$. Here $\mathcal{W}_{j,:,:}$ is a $D \times w$ matrix for the $j$-th output channel. (b) Demonstration of the ConvResNeXt. $f_{1,1} \ldots f_{N,M}$ are the building blocks, each building block is a convolution neural network.

**Definition 2.12.** Let the neural network comprise $N$ residual blocks, each residual block has a parallel architecture with $M$ building blocks, and each building block contains $L$ layers. The number of channels is $w$, and the convolution kernel size is $K$. Given an input $\boldsymbol{x} \in \mathbb{R}^D$, a ConvResNeXt with ReLU activation function can be represented as

$$
\begin{aligned}
f(\boldsymbol{x}) &= \mathbf{W}_{\text{out}} \left( \sum_{m=1}^{M} f_{N,m} + \text{id} \right) \circ \cdots \circ \left( \sum_{m=1}^{M} f_{1,m} + \text{id} \right) \circ P(\boldsymbol{x}), \\
f_{n,m} &= \mathbf{W}_L^{(n,m)} \star \sigma \left( \mathbf{W}_{L-1}^{(n,m)} \star \cdots \star \sigma \left( \mathbf{W}_1^{(n,m)} \star \boldsymbol{x} \right) \right),
\end{aligned}
\tag{5}
$$

where id is the identity operator, $P : \mathbb{R}^D \to \mathbb{R}^{D \times w_0}$ is the padding operator satisfying $P(\boldsymbol{x}) = [\boldsymbol{x}, \ \boldsymbol{0} \ \ldots \ \boldsymbol{0}] \in \mathbb{R}^{D \times w}$, $\{\mathbf{W}_l^{(n,m)}\}_{l=1}^{L}$ is a collection of convolution kernels for $n = 1, \ldots, N, m = 1, \ldots, M$, $\mathbf{W}_{\text{out}} \in \mathbb{R}^{w_L}$ denotes the linear operator for the last layer, and $\star$ is the convolution operation defined in (4).

The structure of ConvResNeXts is shown in Figure 1b. When $M = 1$, the ConvResNeXt defined in (5) reduces to a ConvResNet. For notational simplicity, we omit biases in the neural network structure by extending the input dimension and padding the input with a scalar 1 (See Proposition F.4 for more details). The channel with 0's is used to accumulate the output.

## 3 Theory

In this section, we study a binary classification problem on $\mathcal{M} \subseteq [-1, 1]^D$. Specifically, we are given i.i.d. samples $\{\boldsymbol{x}_i, y_i\}_{i=1}^n \sim \mathcal{D}$ where $\boldsymbol{x}_i \in \mathcal{M}$ and $y_i \in \{0, 1\}$ is the label. The label $y \in \{0, 1\}$ follows the Bernoulli-type distribution

$$
\mathbb{P}(y|\boldsymbol{x}) = \frac{\exp(y f^*(\boldsymbol{x}))}{1 + \exp(f^*(\boldsymbol{x}))}
$$

for some $f^* : \mathcal{M} \to \mathbb{R}$ belonging to the Besov space. More specifically, we make the following assumption on $f^*$.

**Assumption 3.1.** Let $0 < p, q \le \infty$, $d/p < \alpha < \infty$. Assume $f^* \in B_{p,q}^\alpha(\mathcal{M})$ and $\|f^*\|_{B_{p,q}^\alpha(\mathcal{M})} \le C_{\text{F}}$ for some constant $C_{\text{F}} > 0$.

To learn $f^*$, we minimize the empirical logistic risk over the training data:

$$
\hat{f} = \arg \min_{f \in \mathcal{F}^{\text{Conv}}} \frac{1}{n} \sum_{i=1}^{n} \left[ y_i \log(1 + \exp(-f(\boldsymbol{x}_i))) + (1 - y_i) \log(1 + \exp(f(\boldsymbol{x}_i))) \right],
\tag{6}
$$

where $\mathcal{F}^{\text{Conv}}$ is some neural network class specified later. For notational simplicity, we denote the empirical logistic risk function in (6) as $\mathcal{L}_n(f)$, and denote the population logistic risk as

$$
\mathbb{E}_{\mathcal{D}}[\mathcal{L}(f)] = \mathbb{E}_{(\boldsymbol{x},y) \sim \mathcal{D}} \left[ y \log(1 + \exp(-f(\boldsymbol{x}))) + (1 - y) \log(1 + \exp(f(\boldsymbol{x}))) \right].
$$

We next specify the class of ConvResNeXts for learning $f^*$:

$$\mathcal{F}^{\text{Conv}}(N, M, L, K, w, B_{\text{res}}, B_{\text{out}}) = \Big\{ f \mid f \text{ is in the form of (5) with } N \text{ residual blocks. Every}$$

residual block has $M$ building blocks with each building block containing $L$ layers.

Each layer has kernel size bounded by $K$, number of channels bounded by $w$.

$$\sum_{n=1}^{N} \sum_{m=1}^{M} \sum_{\ell=1}^{L} \|\mathbf{W}_\ell^{(n,m)}\|_{\text{F}}^2 \le B_{\text{res}}, \|\mathbf{W}_{\text{out}}\|_{\text{F}}^2 \le B_{\text{out}}, f(\boldsymbol{x}) \in [0,1] \text{ for any } \boldsymbol{x} \in \mathcal{M}. \Big\}. \quad (7)$$

Note that the hyperparameters of $\mathcal{F}^{\text{Conv}}$ will be specified in our theoretical analysis later.

As can be seen, $\mathcal{F}^{\text{Conv}}$ contains the Frobenius norm constraints of the weights. For the sake of computational convenience in practice, such constraints can be replaced with weight decay regularization the residual blocks and the last fully connected layer separately. More specifically, we can use the following alternative formulation:

$$\tilde{f} = \underset{f \in \mathcal{F}^{\text{Conv}}(N,M,L,K,w,\infty,\infty)}{\arg\min} \mathcal{L}_n(f) + \lambda_1 \sum_{n=1}^{N} \sum_{m=1}^{M} \sum_{\ell=1}^{L} \|\mathbf{W}_\ell^{(n,m)}\|_{\text{F}}^2 + \lambda_2 \|\mathbf{W}_{\text{out}}\|_{\text{F}}^2,$$

where $\lambda_1, \lambda_2 > 0$ are properly chosen regularization parameters.

### 3.1 Approximation theory

In this section, we provide a universal approximation theory of ConvResNeXts for Besov functions on a smooth manifold:

**Theorem 3.2.** *For any Besov function $f_0$ on a smooth manifold satisfying $p, q \ge 1, \alpha - d/p > 1$,*

$$\|f_0\|_{B_{p,q}^\alpha(\mathcal{M})} \le C_{\text{F}},$$

*for any $P > 0$ and any ConvResNeXt class $\mathcal{F}^{\text{Conv}}(N, M, L, K, w, B_{\text{res}}, B_{\text{out}})$ satisfying $L = L' + L_0 - 1, L' \ge 3$, where $L_0 = \lceil \frac{D}{K-1} \rceil$, and*

$$MN \ge C_{\mathcal{M}} P, w \ge C_1(dm + D), B_{\text{res}} \le C_2 L/K, B_{\text{out}} \le C_3 C_{\text{F}}^2 ((dm+D)LK)^L (C_{\mathcal{M}} P)^{L-2/p},$$

*there exists $f \in \mathcal{F}^{\text{Conv}}(N, M, L, K, w, B_{\text{res}}, B_{\text{out}})$ such that*

$$\|f - f_0\|_\infty \le C_{\text{F}} C_{\mathcal{M}} \left( C_4 P^{-\alpha/d} + C_5 \exp(-C_6 L' \log P) \right), \quad (8)$$

*where $C_1, C_2, C_3$ are universal constants and $C_4, C_5, C_6$ are constants that only depends on $d$ and $m$, $d$ is the intrinsic dimension of the manifold and $m$ is an integer satisfying $0 < \alpha < \min(m, m - 1 + 1/p)$.*

The approximation error of the network is bounded by the sum of two terms. The first term is a polynomial decay term that decreases with the size of the neural network and represents the trailing term of the B-spline approximation. The second term reflects the approximation error of neural networks to piecewise polynomials, decreasing exponentially with the number of layers. The proof is deferred to Section 4.1 and the appendix.

### 3.2 Estimation theory

**Theorem 3.3.** *Suppose Assumption 3.1 holds. Set $L = L' + L_0 - 1, L' \ge 3$, where $L_0 = \lceil \frac{D}{K-1} \rceil$, and*

$$MN \ge C_{\mathcal{M}} P, \quad P = O(n^{\frac{1-2/L}{2\alpha/d(1-1/L)+1-2/pL}}), \quad w \ge C_1(dm + D).$$

*Let $\hat{f}$ be the global minimizer given in (6) with the function class $\mathcal{F} = \mathcal{F}^{\text{Conv}}(N, M, L, K, w, B_{\text{res}}, B_{\text{out}})$. Then we have*

$$\mathbb{E}_{\mathcal{D}}[\mathcal{L}(\hat{f}(x), y)] - \mathbb{E}_{\mathcal{D}}[\mathcal{L}(f^*(x), y)] \le C_7 \left( \frac{K^{-\frac{2}{L-2}} w^{\frac{3L-4}{L-2}} L^{\frac{3L-2}{L-2}}}{n} \right)^{\frac{\alpha/d(1-2/L)}{2\alpha/d(1-1/L)+1-2/(pL)}}$$
$$+ C_8 \exp(-C_6 L'),$$

*where the logarithmic terms are omitted. $C_1$ is the constant defined in Theorem 3.2, $C_7, C_8$ are constants that depend on $C_{\text{F}}, C_{\mathcal{M}}, d, m$, $K$ is the size of the convolution kernel.*

We would like to make the following remarks about the results:

• **Strong adaptivity:** By setting the width of the neural network to $w = 2C_1D$, the model can adapt to any Besov functions on any smooth manifold, provided that $dm \leq D$. This remarkable flexibility can be achieved simply by tuning the regularization parameter. The cost of overestimating the width is a slight increase in the estimation error.

• **No curse of dimensionality:** the above error rate only depends polynomially on the ambient dimension $D$ and exponentially on the hidden dimension $d$. Since in real data, the hidden dimension $d$ can be much smaller than the ambient dimension $D$, this result shows that neural networks can explore the low-dimension structure of data to overcome the curse of dimensionality.

• **Overparameterization is fine:** the number of building blocks in a ConvResNeXt does not influence the estimation error as long as it is large enough. In other words, this matches the empirical observations that neural networks generalize well despite overparameterization.

• **Close to minimax rate:** The lower bound of the 1-Lipschitz error for any estimator $\theta$ is

$$\min_{\theta} \max_{f^* \in B_{p,q}^\alpha} L(\theta(\mathcal{D}), f^*) \gtrsim n^{-\frac{\alpha/d}{2\alpha/d+1}}.$$

where $\gtrsim$ notation hides a factor of constant. The proof can be found in Appendix E. Comparing to the minimax rate, we can see that as $L \to \infty$, the above error rate converges to the minimax rate up to a constant term. In other words, overparameterized ConvResNeXt can achieve close to the minimax rate in estimating functions in Besov class. In comparison, all kernel ridge regression including any NTKs will have a suboptimal rate lower bounded by $\frac{2\alpha-d}{2\alpha}$, which is suboptimal.

• **Deeper is better:** with larger $L$, the error rate decays faster and gets closer to the minimax rate. This indicates that deeper model can achieve better performance than shallower models.

• **Tradeoff between width and depth:** With a fixed budget in the number of parameters, the tradeoff between width and depth is crucial for achieving the best performance, and this often requires repeated, time-consuming experiments. On the other hand, our results suggests that such a tradeoff less important in a ResNeXt. The lower bound of error does not depend on the arrangements of the residual blocks $M$ and $N$, as long as their product is large enough. This can partly explain the benefit of ResNeXt over other architecture.

By choosing $L = O(\log(n))$ in Theorem 3.3, the second term in the error can be merged with the first term, and close to the minimax rate can be achieved:

**Corollary 3.4.** *Given the conditions in Theorem 3.3, set the depth of each block is $L = O(\log(n))$ and then the estimation error of the empirical risk minimizer $\hat{f}$ satisfies*

$$\mathbb{E}_{\mathcal{D}}[\mathcal{L}(\hat{f}(\boldsymbol{x}), y)] \leq \mathbb{E}_{\mathcal{D}}[\mathcal{L}(f^*)] + \tilde{O}(n^{-\frac{\alpha/d}{2\alpha/d+1}(1-o(1))}),$$

*where $\tilde{O}(\cdot)$ omits the logarithmic term.*

The proof of Theorem 3.3 is deferred to Section 4.2 and Section D.2. The key technique is computing the critical radius of the local Gaussian complexity by bounding the covering number of weight-decayed ConvResNeXts. This technique provides a tighter bound than choosing a single radius of the covering number as in Suzuki [32], Zhang and Wang [43]. The covering number of an overparameterized ConvResNeXt with norm constraint (Lemma 4.1) is one of our key contributions.

## 4 Proof overview

### 4.1 Approximation error

We follow the method in Liu et al. [25] to construct a neural network that achieves the approximation error we claim. It is divided into the following steps:

• **Step 1: Decompose the target function into the sum of locally supported functions.**

In this work, we adopt a similar approach to [25] and partition $\mathcal{M}$ using a finite number of open balls on $\mathbb{R}^D$. Specifically, we define $B(\boldsymbol{c}_i, r)$ as the set of unit balls with center $\boldsymbol{c}_i$ and radius $r$ such that their union covers the manifold of interest, i.e., $\mathcal{M} \subseteq \cup_{i=1}^{C_{\mathcal{M}}} B(\boldsymbol{c}_i, r)$. This allows us to partition

the manifold into subregions $U_i = B(\boldsymbol{c}_i, r) \cap \mathcal{M}$, and further decompose a smooth function on the manifold into the sum of locally supported smooth functions with linear projections. The existence of function decomposition is guaranteed by the existence of partition of unity stated in Proposition 2.6. See Section C.1 for the detail.

• **Step 2: Locally approximate the decomposed functions using cardinal B-spline basis functions.** In the second step, we decompose the locally supported Besov functions achieved in the first step using B-spline basis functions. The existence of the decomposition was proven by Dũng [11], and was applied in a series of works [43, 32, 25]. The difference between our result and previous work is that we define a norm on the coefficients and bound this norm, instead of bounding the maximum value. The detail is deferred to Section C.2.

• **Step 3: Approximate the polynomial functions using neural networks.** In this section, we follow the method in Zhang and Wang [43], Suzuki [32], Liu et al. [25] and show that neural networks can be used to approximate polynomial functions, including B-spline basis functions and the distance function. The key technique is to use a neural network to approximate square function and multiply function [3]. The detail is deferred to the appendix. Specifically, Lemma F.3 proves that a neural network with width $w = O(dm)$ and depth $L$ can approximate B-spline basis functions, and the error decreases exponentially with $L$; Similarly, Proposition C.3 shows that a neural network with width $w = O(D)$ can approximately calculate the distance between two points $d^2(\boldsymbol{x}; \boldsymbol{c})$, with precision decreasing exponentially with the depth.

• **Step 4: Use a ConvResNeXt to Approximate the target function.** Using the results above, the target function can be (approximately) decomposed as

$$\sum_{i=1}^{C_{\mathcal{M}}} \sum_{j=1}^{P} a_{i,k_j,\boldsymbol{s}_j} M_{m,k_j,\boldsymbol{s}_j} \circ \phi_i \times \mathbf{1}(\boldsymbol{x} \in B(\boldsymbol{c}_i, r)). \tag{9}$$

We first demonstrate that a ReLU neural network taking two scalars $a, b$ as the input, denoted as $a \tilde{\times} b$, can approximate $y \times \mathbf{1}(\boldsymbol{x} \in B_{r,i})$, where $\tilde{\times}$ satisfy that $y \tilde{\times} 1 = y$ for all $y$, and $y \tilde{\times} \tilde{x} = 0$ if any of $x$ or $y$ is 0, and the soft indicator function $\tilde{\mathbf{1}}(\boldsymbol{x} \in B_{r,i})$ satisfy $\tilde{\mathbf{1}}(\boldsymbol{x} \in B_{r,i}) = 1$ when $x \in B_{r,i}$, and $\tilde{\mathbf{1}}(\boldsymbol{x} \in B_{r,i}) = 0$ when $x \notin B_{r+\Delta,i}$. The detail is deferred to Section C.3.

Then, we show that it is possible to construct $MN = C_{\mathcal{M}}P$ number of building blocks, such that each building block is a feedforward neural network with width $C_1(md + D)$ and depth $L$, where $m$ is an integer satisfying $0 < \alpha < min(m, m - 1 + 1/p)$. The $k$-th building block (the position of the block does not matter) approximates $a_{i,k_j,\boldsymbol{s}_j} M_{m,k_j,\boldsymbol{s}_j} \circ \phi_i \times \mathbf{1}(\boldsymbol{x} \in B(\boldsymbol{c}_i, r))$, where $i = ceiling(k/N), j = rem(k, N)$. Each building block has where a sub-block with width $D$ and depth $L - 1$ approximates the chart selection, a sub-block with width $md$ and depth $L - 1$ approximates the B-spline function, and the last layer approximates the multiply function. The norm of this block is bounded by

$$\sum_{\ell=1}^{L} \|\mathbf{W}_\ell^{(i,j)}\|_{\mathrm{F}}^2 \leq O(2^{2k/L} dmL + DL). \tag{10}$$

Making use of the 1-homogeneous property of the ReLU function, by scaling all the weights in the neural network, these building blocks can be combined into a neural network with residual connections, that approximate the target function and satisfy our constraint on the norm of weights. See Section C.4 for the detail.

By applying Lemma C.6, which shows that any $L$-layer feedforward neural network can be reformulated as an $L + L_0 - 1$-layer convolution neural network, the neural network constructed above can be converted into a ConvResNeXt that satisfies the conditions in Theorem 3.2.

## 4.2 Estimation error

We first prove the covering number of an overparameterized ConvResNeXt with norm-constraint as in Lemma 4.1, then compute the critical radius of this function class using the covering number as in Corollary F.5. The critical radius can be used to bound the estimation error as in Theorem 14.20 in Wainwright [36]. The proof is deferred to Section D.2.

**Lemma 4.1.** *Consider a neural network defined in Definition 2.12. Let the last layer of this neural network is a single linear layer with norm $\|W_{\text{out}}\|_{\mathrm{F}}^2 \leq B_{\text{out}}$. Let the input of this neural network*

*satisfy* $\|\boldsymbol{x}\|_2 \leq 1, \forall x$, *and is concatenated with 1 before feeding into this neural network so that part of the weight plays the role of the bias. The covering number of this neural network is bounded by*

$$\log \mathcal{N}(\cdot, \delta) \lesssim w^2 L B_{\mathrm{res}}^{\frac{1}{1-2/L}} K^{\frac{2-2/L}{1-2/L}} \cdot \left(B_{\mathrm{out}}^{1/2} \exp((KB_{\mathrm{res}}/L)^{L/2})\right)^{\frac{2/L}{1-2/L}} \delta^{-\frac{2/L}{1-2/L}}, \quad (11)$$

*where the logarithmic term is omitted.*

The key idea of the proof is to split the building block into two types ("small blocks" and "large blocks") depending on whether the total norm of the weights in the building block is smaller than $\epsilon$. By properly choosing $\epsilon$, we prove that if all the "small blocks" in this neural network are removed, the perturbation to the output for any input $\|x\| \leq 1$ is no more than $\delta/2$, so the covering number of the ConvResNeXt is only determined by the number of "large blocks", which is no more than $B_{\mathrm{res}}/\epsilon$.

*Proof.* Using the inequality of arithmetic and geometric means, from Proposition F.6, Proposition F.8 and Proposition F.9, if any residual block is removed, the perturbation to the output is no more than

$$p_m := (KB_m/L)^{L/2} B_{\mathrm{out}}^{1/2} \exp((KB_{\mathrm{res}}/L)^{L/2}),$$

where $B_m$ is the total norm of parameters in this block. Because of that, the residual blocks can be divided into two kinds depending on the norm of the weights $B_m < \epsilon$ ("small blocks") and $B_m \geq \epsilon$ ("large blocks"). If all the "small blocks" are removed, the perturbation to the output for any input $\|\boldsymbol{x}\|_2 \leq 1$ is no more than

$$\sum_{m:B_m < \epsilon} p_m \leq \exp((KB_{\mathrm{res}}/L)^{L/2}) K^{L/2} B_{\mathrm{res}} B_{\mathrm{out}}^{1/2} (\epsilon/L)^{L/2-1}/L.$$

Choosing $\epsilon = L\left(\frac{\delta L}{2 \exp((KB_{\mathrm{res}}/L)^{L/2}) K^{L/2} B_{\mathrm{res}} B_{\mathrm{out}}^{1/2}}\right)^{1/(L/2-1)}$, the perturbation above is no more than $\delta/2$. The covering number can be determined by the number of the "large blocks" in the neural network, which is no more than $B_{\mathrm{res}}/\epsilon$. As for any block, $B_{\mathrm{in}} L_{\mathrm{post}} \leq B_{\mathrm{out}}^{1/2} \exp((KB_{\mathrm{res}}/L)^{L/2})$, taking our chosen $\epsilon$ finishes the proof, where $B_{\mathrm{in}}$ is the upper bound of the input to this block defined in Proposition D.1, and $L_{\mathrm{post}}$ is the Lipschitz constant of all the layers following the block.

$\square$

*Remark* 4.2. The proof of Lemma 4.1 shows that under weight decay, the building blocks in a ConvResNeXt are sparse, i.e. only a finite number of blocks contribute non-trivially to the network even though the model can be overparameterized. This explains why a ConvResNeXt can generalize well despite overparameterization, and provide a new perspective in explaining why residual connections improve the performance of deep neural networks.

## 5   Discussions

This paper focuses on developing insightful generalization bounds for the regularized empirical risk minimizer. We opt not to delve into the end-to-end analysis of optimization algorithms in order to explore the adaptivity of complex architectures such as ConvResNeXts, while works on optimization behaviour of neural networks are limited to simple network structures [28, 39]. Notably, this approach to decouple learning and optimization has been widely adopted [3, 17, 31, 6, 25]. We made the same choice in the interest of getting a more fine-grained learning theory. However, our paper considers weight decay and overparameterization which are tightly connected to real-world training of neural networks, and can be the most promising work to bridge the gap between optimization and statistical guarantees. We defer more details to the appendix, including discussions on the Besov space and numerical experiments for supporting our theories as well as supplementary technical proof.

## Acknowledgments

The work was partially supported by NSF Award No 2134214. KZ and YW were with UCSB and MC was with Princeton when the work was completed. YT contributed to the project during his summer 2023 visit to UCSB. We appreciate anonymous reviewers and ACs for their input.

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

# A  Why Besov Classes?

In this section, we discuss why we choose to consider the Besov class of functions and why this makes our results particularly interesting.

To see this, we need to first define two smaller function classes: the Holder class and the Sobolev class. Instead of giving fully general definitions for these function classes let us illustrate their main differences using univariate functions defined on $[0, 1]$. We also introduce the so-called Total Variation class — which is sandwiched in between $\text{Besov}(p = 1, q = 1)$ and $\text{Besov}(p = 1, q = \infty)$.

- Holder class functions satisfy $|f^{(\alpha)}(x)| < C$ for all $x$.
- Sobolev class functions satisfy $\int_{[0,1]} |f^{(\alpha)}(x)|^2 dx < C$
- Total Variation class functions satisfy $\int_{[0,1]} |f^{(\alpha)}(x)| dx < C$

The L1-norm used in defining total variation class makes it the most flexible of the three. It allows functions with $\alpha^{th}$ order derivative $f^{(\alpha)}(x)$ to be very large at some places, e.g., Dirac delta functions, while Holder and Sobolev class functions cannot contain such spikes (no longer integrable in Sobolev norm above).

Generically speaking under the appropriate scaling: **Holder $\subset$ Sobolev $\subset$ Besov**. The Besov space contains functions with heterogeneous smoothness while Holder and Sobolev classes contain functions with homogeneous smoothness. Despite the Besov space being larger, it has the same minimax rate of $n^{-(2\alpha)/(2\alpha+d)}$ as the smaller Holder and Sobolev class.

**A new perspective on overparameterized NN.** We study the adaptivity of deep networks in overparameterized regimes. The most popular method for understanding overparameterization is through the neural tangent kernel (NTK) regime. However, based on the classical linear smoother lower-bound for estimating functions in Besov classes with $p = 1$ [9, 10], all kernel ridge regression including any NTKs will have a suboptimal rate lower bounded by $n^{-\frac{2\alpha-d}{2\alpha}}$. To say it differently, there is a formal separation between NTKs and the optimal method. The same separation does not exist in smaller function classes such as Sobolev and Holders because they are more homogeneously smooth.

In summary, in order to study what neural networks can achieve that is not achievable by kernels, e.g., NTK; we had to define and approximate Besov class functions. Our results show that ConvResNeXT not only overcomes the curse of dimensionality of the ambient space, but also has nearly optimal dependence in the intrinsic dimension $d$ — in contrast to the kernel-based approaches.

We believe this offers a new perspective to understand overparameterization and is more fine-grained that of NTK.

# B  Numerical Simulation

In this section, we validate our theoretical findings with numerical experiments. We focus on nonparametric regression problems for simplicity and consider the following function $f_0 : \mathbb{R}^D \to \mathbb{R}$:

$$f_0(x) = \tilde{f}_0(Ux) = \tilde{f}_0(\tilde{x})$$

where $U \in \mathbb{R}^{D \times D}$ is a randomly-chosen rotation matrix and $\tilde{x} = Ux \in \mathbb{R}^D$ satisfies that for $t \in [0, 1]$, the first three coordinates

$$\tilde{x}_1 = t \sin(4\pi t), \quad \tilde{x}_2 = t \cos(4\pi t), \quad \tilde{x}_3 = t(1 - t),$$

and the remaining coordinates of $\tilde{x}$ are irrelevant features iid sampled from a uniform distribution. Note that the first three coordinates of $\tilde{x}$ are completely determined by a scalar $t$, and the corresponding label $y$ is determined by $t$ via a piecewise linear function, i.e., for a bag of $t_1, ..., t_n \in [0, 1]$, we can generate a labeled dataset by $y_i = g_0(t_i) + \mathcal{N}(0, 1)$. An illustration of the function $f_0$ is given in Figure 2 where colors indicate the value.

**Role of irrelevant features and rotation.** The purpose of irrelevant features and rotation is to make the problem harder and more interesting.

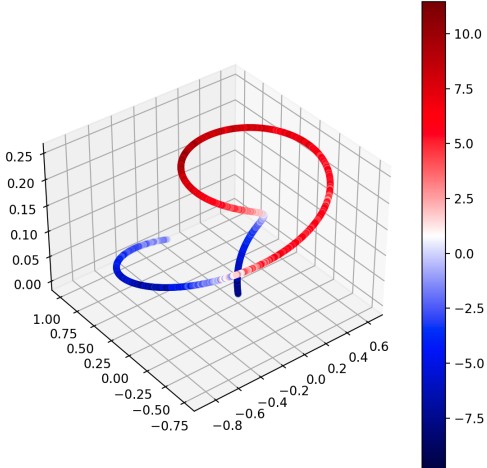

Figure 2: Illustration of a Besov function on 1-dimensional manifold embedded in a 3-dimensional ambient space.

$$x_{i,1} = t_i \sin(4\pi t_i), \quad x_{i,2} = t_i \cos(4\pi t_i), \quad x_{i,3} = t_i(1 - t_i),$$

where $t_i, i = 1, \ldots, n$ are evenly spaced over $[0, 1]$. This process generates a 1-dimensional manifold in $\mathbb{R}^3$ which does not intersect with itself, as shown in Figure 2.

**Baseline methods** To estimate the underlying function on a manifold, we conducted experiments with ConvResNeXts (this paper), as well as a mix of popular off-the-shelf methods including kernel ridge regression, XGBoost, Decision tree, Lasso regression, and Gaussian Processes.

**Hyperparameter choices.** In all the experiments the following architecture was used for PNN: $w = 6$, $L = 10$, $M = 4$, batch_size $= 128$, learning_rate $= 1e - 3$

In all the experiments the following architecture was used for ConvResNeXt: $w = 8$, $L = 6$, $K = 6$, $M = 2$, $N = 2$. Batch_size and learning_rate were adjusted for each task.

For off-the-shelf methods, their hyperparameters are either tuned automatically or avoided using tools provided from the package, e.g., GP. For GP, a Matern kernel is used, and for ridge regression, the standard Gaussian RBF kernel is used.

**Results.** Our results are reported in Figure 3, 4, 5 which reports the *mean square error* (MSE) as a function of the effective degree-of-freedom of each method, ambient dimension $D$ and also the number of data points $n$ respectively.

As we can see in Figure 3, ConvResNeXt is able to achieve the lowest MSE at a relatively smaller degree of freedom. It outperforms the competing methods with notable margins despite using a simpler hypothesis.

Figure 4 illustrates that standard non-parametric methods such as kernel ridge regression and Gaussian processes deteriorate quickly as the ambient dimension gets bigger. On the contrary, ConvResNeXt and PNN obtain results that are almost dimension-independent due to the representation learning that helps identify the low-dimensional manifold.

Finally, the log-log plot in Figure 5 demonstrates that there is a substantially different rate of convergence between our methods and kernel ridge regression and GPs, indicating the same formal separation that we have established in the theoretical part — kernels must be suboptimal for estimating Besov classes while the neural architectures we considered can be locally adaptive and nearly optimal for Besov classes.

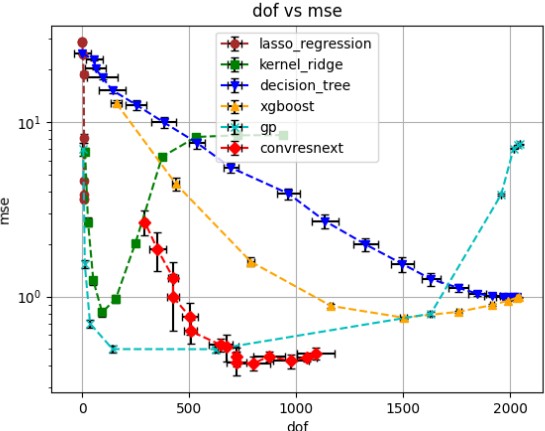

Figure 3: MSE as a function of the effective degree of freedom (dof) of different methods.

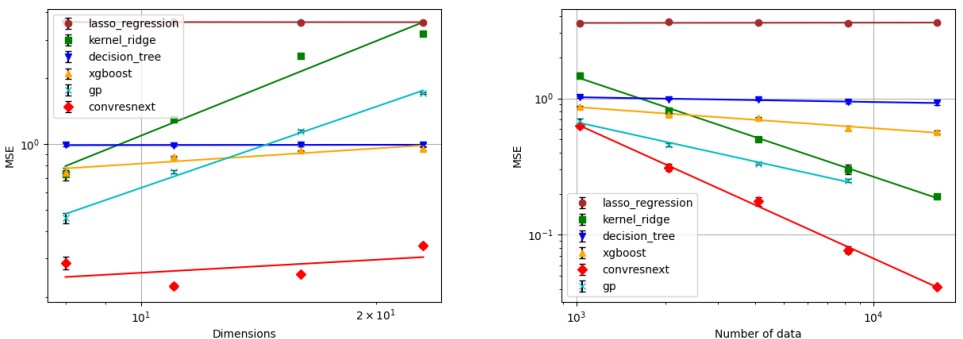

Figure 4: MSE as a function of dimension $D$.    Figure 5: MSE as function of sample size $n$.

## C  Proof of the approximation theory

### C.1  Decompose the target function into the sum of locally supported functions.

**Lemma C.1.** *Approximating Besov function on a smooth manifold using B-spline: Let $f \in B_{p,q}^{\alpha}(\mathcal{M})$. There exists a decomposition of $f$:*

$$f(\boldsymbol{x}) = \sum_{i=1}^{C_{\mathcal{M}}} \tilde{f}_i \circ \phi_i(\boldsymbol{x}) \times \mathbf{1}(\boldsymbol{x} \in B(\boldsymbol{c}_i, r)),$$

*and $\tilde{f}_i = f \cdot \rho_i \in B_{p,q}^{\alpha}$, $\sum_{i=1}^{C_{\mathcal{M}}} \|\tilde{f}_i\|_{B_{p,q}^{\alpha}} \le C\|f\|_{B_{p,q}^{\alpha}(\mathcal{M})}$, $\phi_i : \mathcal{M} \to \mathbb{R}^d$ are linear projections, $B(\boldsymbol{c}_i, r)$ denotes the unit ball with radius $r$ and center $\boldsymbol{c}_i$.*

The lemma is inferred by the existence of the partition of unity, which is given in Proposition 2.6.

### C.2  Locally approximate the decomposed functions using cardinal B-spline basis functions.

**Proposition C.2.** *For any function in the Besov space on a compact smooth manifold $f^* \in B_{p,q}^{s}(\mathcal{M})$, any $N \ge 0$, there exists an approximated to $f^*$ using cardinal B-spline basis functions:*

$$\tilde{f} = \sum_{i=1}^{C_{\mathcal{M}}} \sum_{j=1}^{P} a_{i,k_j,\boldsymbol{s}_j} M_{m,k_j,\boldsymbol{s}_j} \circ \phi_i \times \mathbf{1}(\boldsymbol{x} \in B(\boldsymbol{c}_i, r)),$$

*where $m$ is the integer satisfying $0 < \alpha < min(m, m - 1 + 1/p)$, $M_{m,k,s} = M_m(2^k(\cdot - s))$, $M_m$ denotes the B-spline basis function defined in (2), the approximation error is bounded by*

$$\|f - \tilde{f}\|_\infty \le C_9 C_{\mathcal{M}} P^{-\alpha/d}$$

*and the coefficients satisfy*

$$\|\{2^{k_j} a_{i,k_j,s_j}\}_{i,j}\|_p \le C_{10}\|f\|_{B^\alpha_{p,q}(\mathcal{M})}$$

*for some constant $C_9, C_{10}$ that only depends on $\alpha$.*

As will be shown below, the scaled coefficients $2^{k_j} a_{i,k_j,s_j}$ corresponds to the total norm of the parameters in the neural network to approximate the B-spline basis function, so this lemma is the key to get the bound of norm of parameters in (12).

*Proof.* From the definition of $B^\alpha_{p,q}(\mathcal{M})$, and applying Proposition 2.6, there exists a decomposition of $f^*$ as

$$f^* = \sum_{i=1}^{C_{\mathcal{M}}}(f_i) = \sum_{i=1}^{C_{\mathcal{M}}}(f_i \circ \phi_i^{-1}) \circ \phi_i \times \mathbf{1}_{U_i},$$

where $f_i := f^* \cdot \rho_i$, $\rho_i$ satisfy the condition in Definition 2.5, and $f_i \circ \phi_i^{-1} \in B^\alpha_{p,q}$. Using Proposition F.2, for any $i$, one can approximate $f_i \circ \phi_i^{-1}$ with $\bar{f}_i$:

$$\bar{f}_i = \sum_{j=1}^{P} a_{i,k_j,s_j} M_{m,k_j,s_j}$$

such that $\|f_i \circ \phi_i^{-1}\|_\infty \le C_1 M^{-\alpha/d}$, and the coefficients satisfy

$$\|\{2^{k_j} a_{k_j,s_j}\}_j\|_p \le C_{10}\|f_i \circ \phi_i^{-1}\|_{B^\alpha_{p,q}}.$$

Define

$$\bar{f} = \sum_{i=1}^{C_{\mathcal{M}}} \bar{f}_i \circ \phi_i \times \mathbf{1}_{U_i}.$$

one can verify that $\|f - \tilde{f}\|_\infty \le C_9 C_{\mathcal{M}} N^{-\alpha/d}$. On the other hand, using triangular inequality (and padding the vectors with 0),

$$\|\{2^{k_j} a_{i,k_j,s_j}\}_{i,j}\|_p \le \sum_{i=1}^{C_{\mathcal{M}}}\|\{2^{k_j} a_{i,k_j,s_j}\}_j\|_p \le \sum_{i=1}^{C_{\mathcal{M}}} C_{10}\|f_i \circ \phi_i^{-1}\|_{B^\alpha_{p,q}} = C_{10}\|f^*\|_{B^\alpha_{p,q}(\mathcal{M})},$$

which finishes the proof.

$\square$

### C.3 Neural network for chart selection

In this section, we demonstrate that a feedforward neural network can approximate the chart selection function $z \times \mathbf{1}(\boldsymbol{x} \in B(\boldsymbol{c}_i, r))$, and it is error-free as long as $z = 0$ when $r < d(\boldsymbol{x}, \boldsymbol{c}_i) < R$. We start by proving the following supporting lemma:

**Proposition C.3.** *Fix some constant $B > 0$. For any $\boldsymbol{x}, \boldsymbol{c} \in \mathrm{R}^D$ satisfying $|x_i| \le B$ and $|c_i| \le B$ for $i = 1, \ldots, D$, there exists an $L$-layer neural network $\tilde{d}(\boldsymbol{x}; \boldsymbol{c})$ with width $w = O(d)$ that approximates $d^2(\boldsymbol{x}; \boldsymbol{c}) = \sum_{i=1}^{D}(x_i - c_i)^2$ such that $|\tilde{d}^2(\boldsymbol{x}; \boldsymbol{c}) - d^2(\boldsymbol{x}; \boldsymbol{c})| \le 8DB^2 \exp(-C_{11}L)$ with an absolute constant $C_{11} > 0$ when $d(\boldsymbol{x}; \boldsymbol{c}) < \tau$, and $\tilde{d}^2(\boldsymbol{x}; \boldsymbol{c}) \ge \tau^2$ when $d(\boldsymbol{x}; \boldsymbol{c}) \ge \tau$, and the norm of the neural network is bounded by*

$$\sum_{\ell=1}^{L}\|W_\ell\|_{\mathrm{F}}^2 + \|b_\ell\|_2^2 \le C_{12}DL.$$

*Proof.* The proof is given by construction. By Proposition 2 in Yarotsky(2017), the function $f(x) = x^2$ on the segment $[0, 2B]$ can be approximated with any error $\epsilon > 0$ by a ReLU network $g$ having depth and the number of neurons and weight parameters no more than $c \log(4B^2/\epsilon)$ with an absolute constant $c$. The width of the network $g$ is an absolute constant. We also consider a single layer ReLU neural network $h(t) = \sigma(t) - \sigma(-t)$, which is equal to the absolute value of the input.

Now we consider a neural network $G(\boldsymbol{x}; \boldsymbol{c}) = \sum_{i=1}^{D} g \circ h(x_i - c_i)$. Then for any $\boldsymbol{x}, \boldsymbol{c} \in \mathrm{R}^D$ satisfying $|x_i| \leq B$ and $|c_i| \leq B$ for $i = 1, \ldots, D$, we have

$$
\begin{aligned}
|G(\boldsymbol{x}; \boldsymbol{c}) - d^2(\boldsymbol{x}; \boldsymbol{c})| &\leq \left| \sum_{i=1}^{D} g \circ h(x_i - c_i) - \sum_{i=1}^{D} (x_i - c_i)^2 \right| \\
&\leq \sum_{i=1}^{D} \left| g \circ h(x_i - c_i) - (x_i - c_i)^2 \right| \\
&\leq D\epsilon.
\end{aligned}
$$

Moreover, define another neural network

$$
\begin{aligned}
F(\boldsymbol{x}; \boldsymbol{c}) &= -\sigma(\tau^2 - D\epsilon - G(\boldsymbol{x}; \boldsymbol{c})) + \tau^2 \\
&= \begin{cases} G(\boldsymbol{x}; \boldsymbol{c}) + D\epsilon & \text{if } G(\boldsymbol{x}; \boldsymbol{c}) < \tau^2 - D\epsilon, \\ \tau^2 & \text{if } G(\boldsymbol{x}; \boldsymbol{c}) \geq \tau^2 - D\epsilon, \end{cases}
\end{aligned}
$$

which has depth and the number of neurons no more than $c' \log(4B^2/\epsilon)$ with an absolute constant $c'$. The weight parameters of $G$ are upper bounded by $\max\{\tau^2, D\epsilon, c \log(4B^2/\epsilon)\}$ and the width of $G$ is $O(D)$.

If $d^2(\boldsymbol{x}; \boldsymbol{c}) < \tau^2$, we have

$$
\begin{aligned}
|F(\boldsymbol{x}; \boldsymbol{c}) - d^2(\boldsymbol{x}; \boldsymbol{c})| &= |-\sigma(\tau^2 - D\epsilon - G(\boldsymbol{x}; \boldsymbol{c})) + \tau^2 - d^2(\boldsymbol{x}; \boldsymbol{c})| \\
&= \begin{cases} |G(\boldsymbol{x}; \boldsymbol{c}) - d^2(\boldsymbol{x}; \boldsymbol{c}) + D\epsilon| & \text{if } G(\boldsymbol{x}; \boldsymbol{c}) < \tau^2 - D\epsilon, \\ \tau^2 - d^2(\boldsymbol{x}; \boldsymbol{c}) & \text{if } G(\boldsymbol{x}; \boldsymbol{c}) \geq \tau^2 - D\epsilon. \end{cases}
\end{aligned}
$$

For the first case when $G(\boldsymbol{x}; \boldsymbol{c}) < \tau^2 - D\epsilon$, $|F(\boldsymbol{x}; \boldsymbol{c}) - d^2(\boldsymbol{x}; \boldsymbol{c})| \leq 2D\epsilon$ since $d^2(\boldsymbol{x}; \boldsymbol{c})$ can be approximated by $G(\boldsymbol{x}; \boldsymbol{c})$ up to an error $\epsilon$. For the second case when $G(\boldsymbol{x}; \boldsymbol{c}) \geq \tau^2 - D\epsilon$, we have $d^2(\boldsymbol{x}; \boldsymbol{c}) \geq G(\boldsymbol{x}; \boldsymbol{c}) - D\epsilon \geq \tau^2 - 2D\epsilon$ and . Thereby we also have $|F(\boldsymbol{x}; \boldsymbol{c}) - d^2(\boldsymbol{x}; \boldsymbol{c})| \leq 2D\epsilon$.

If $d^2(\boldsymbol{x}; \boldsymbol{c}) \geq \tau^2$ instead, we will obtain $G(\boldsymbol{x}; \boldsymbol{c}) \geq d^2(\boldsymbol{x}; \boldsymbol{c}) - D\epsilon \geq \tau^2 - D\epsilon$. This gives that $F(\boldsymbol{x}; \boldsymbol{c}) = \tau^2$ in this case.

Finally, we take $\epsilon = 4B^2 \exp(-L/c')$. Then $F(\boldsymbol{x}; \boldsymbol{c})$ is an $L$-layer neural network with $O(L)$ neurons. The weight parameters of $G$ are upper bounded by $\max\{\tau^2, 4DB^2 \exp(-L/c'), cL/c'\}$ and the width of $G$ is $O(D)$. Moreover, $F(\boldsymbol{x}; \boldsymbol{c})$ satisfies $|F(\boldsymbol{x}; \boldsymbol{c}) - d^2(\boldsymbol{x}; \boldsymbol{c})| < 8DB^2 \exp(-L/c')$ if $d^2(\boldsymbol{x}; \boldsymbol{c}) \leq \tau^2$ and $F(\boldsymbol{x}; \boldsymbol{c}) = \tau^2$ if $d^2(\boldsymbol{x}; \boldsymbol{c}) \geq \tau^2$. □

**Proposition C.4.** *There exists a single layer ReLU neural network that approximates $\tilde{\times}$, such that for all $0 \leq x \leq C, y \in \{0, 1\}$, $x \tilde{\times} y = x$ when $y = 1$, and $x \tilde{\times} y = 0$ when either $x = 0$ or $y = 0$.*

*Proof.* Consider a single layer neural network $g(x, y) := A_2 \sigma(A_1(x, y)^\top)$ with no bias, where

$$
A_1 = \begin{bmatrix} -\frac{1}{C} & 1 \\ 0 & 1 \end{bmatrix}, \quad A_2 = \begin{bmatrix} -C \\ C \end{bmatrix}.
$$

Then we can rewrite the neural network $g$ as $g(x, y) = -C\sigma(-x/C + y) + C\sigma(y)$. If $y = 1$, we will have $g(x, y) = -C\sigma(-x/C + 1) + C = x$, since $x \leq C$. If $y = 0$, we will have $g(x, y) = -C\sigma(-x/C) = 0$, since $x \geq 0$. Thereby we can conclude the proof. □

By adding a single linear layer

$$
y = \frac{1}{R - r - 2\Delta}(\sigma(R - \Delta - x) - \sigma(r + \Delta - x))
$$

after the one shown in Proposition C.3, where $\Delta = 8DB^2 \exp(-CL)$ denotes the error in Proposition C.3, one can approximate the indicator function $\mathbf{1}(\boldsymbol{x} \in B(\boldsymbol{c}_i, r))$ such that it is error-free when $d(\boldsymbol{x}, \boldsymbol{c}_i) \leq r$ or $\geq R$. Choosing $R \leq \tau/2, r < R - 2\Delta$, and combining with Proposition C.4, the proof is finished. Considering that $f_i$ is locally supported on $B(\boldsymbol{c}_i, r)$ for all $i$ by our method of construction, the chart selection part does not incur any error in the output.

## C.4 Constructing the neural network to Approximate the target function

In this section, we focus on the neural network with the same architecture as a ResNeXt in Definition 2.12 but replacing each building block with a feedforward neural network, and prove that it can achieve the same approximation error as in Theorem 3.2. For technical simplicity, we assume that the target function $f^* \in [0, 1]$ without loss of generality. Then our analysis automatically holds for any bounded function.

**Theorem C.5.** *For any $f^*$ under the same condition as Theorem 3.2, any neural network architecture with residual connections containing $N$ number of residual blocks and each residual block contains $M$ number of feedforward neural networks in parallel, where the depth of each feedforward neural networks is $L$, width is $w$:*

$$f = \mathbf{W}_{\text{out}} \cdot \left(1 + \sum_{m=1}^{M} f_{N,m}\right) \circ \cdots \circ \left(1 + \sum_{m=1}^{M} f_{1,m}\right)$$

$$f_{n,m} = \mathbf{W}_L^{(n,m)} \sigma(\mathbf{W}_{L-1}^{(n,m)} \ldots \sigma(\mathbf{W}_1^{(n,m)} \boldsymbol{x})) \circ P(\boldsymbol{x}),$$

*where $P(\boldsymbol{x}) = [\boldsymbol{x}^T, 1, 0]^T$ is the padding operation,*

*satisfying*

$$MN \geq C_{\mathcal{M}} P, \quad w \geq C_1(dm + D),$$

$$B_{\text{res}} := \sum_{n=1}^{N} \sum_{m=1}^{M} \sum_{\ell=1}^{L} \|\mathbf{W}_{\ell}^{(n,m)}\|_{\text{F}}^2 \leq C_2 L, \tag{12}$$

$$B_{\text{out}} := \|\mathbf{W}_{\text{out}}\|_{\text{F}}^2 \leq C_3 C_{\text{F}}^2 ((dm + D)L)^L (C_{\mathcal{M}} P)^{L-2/p},$$

*there exists an instance $f$ of this ResNeXt class, such that*

$$\|f - f^*\|_{\infty} \leq C_{\text{F}} C_{\mathcal{M}} \left(C_4 P^{-\alpha/d} + C_5 \exp(-C_6 L \log P)\right), \tag{13}$$

*where $C_1, C_2, C_3, C_4, C_5, C_6$ are the same constants as in Theorem 3.2.*

*Proof.* We first construct a parallel neural network to approximate the target function, then scale the weights to meet the norm constraint while keeping the model equivalent to the one constructed in the first step, and finally transform this parallel neural network into the ConvResNeXt as claimed.

Combining Lemma F.3, Proposition C.3 and Proposition C.4, by putting the neural network in Lemma F.3 and Proposition C.3 in parallel and adding the one in Proposition C.4 after them, one can construct a feedforward neural network with bias with depth $L$, width $w = O(d) + O(D) = O(d)$, that approximates $M_{m,k_j,\boldsymbol{s}_j}(\boldsymbol{x}) \times \mathbf{1}(\boldsymbol{x} \in B(\boldsymbol{c}_i, r))$ for any $i, j$.

To construct the neural network with residual connections that approximates $f^*$, we follow the method in Oono and Suzuki [30], Liu et al. [25]. This network uses separate channels for the inputs and outputs. Let the input to one residual layer be $[\boldsymbol{x}_1, y_1]$, the output is $[\boldsymbol{x}_1, y_1 + f(x_1)]$. As a result, if one scale the outputs of all the building blocks by any scalar $a$, then the last channel of the output of the entire network is also scaled by $a$. This property allows us to scale the weights in each building block while keeping the model equivalent. To compensate for the bias term, Proposition F.4 can be applied. This only increases the total norm of each building block by no larger than a constant term that depends only $L$, which is no more than a factor of constant.

Let the neural network constructed above has parameter $\tilde{\mathbf{W}}_1^{(i,j)}, \tilde{\boldsymbol{b}}_1^{(i,j)}, \ldots, \tilde{\mathbf{W}}_L^{(i,j)}, \boldsymbol{b}_L^{(i,j)}$ in each layer, one can construct a building block without bias as

$$\mathbf{W}_1^{(i,j)} = \begin{bmatrix} \tilde{\mathbf{W}}_1^{(i,j)} & \tilde{\boldsymbol{b}}_1^{(i,j)} & 0 \\ 0 & 1 & 0 \end{bmatrix}, \quad \mathbf{W}_{\ell}^{(i,j)} = \begin{bmatrix} \tilde{\mathbf{W}}_{\ell}^{(i,j)} & \tilde{\boldsymbol{b}}_{\ell}^{(i,j)} \\ 0 & 1 \end{bmatrix} \quad \mathbf{W}_L^{(i,j)} = \begin{bmatrix} 0 & 0 \\ 0 & 0 \\ \tilde{\mathbf{W}}_L^{(i,j)} & \tilde{\boldsymbol{b}}_L^{(i,j)} \end{bmatrix}.$$

Remind that the input is padded with the scalar 1 before feeding into the neural network, the above construction provide an equivalent representation to the neural network including the bias, and route the output to the last channel. From Lemma F.3, it can be seen that the total square norm of this block is bounded by (10).

Finally, we scale the weights in the each block, including the "1" terms to meet the norm constraint. Thanks to the 1-homogeneous property of ReLU layer, and considering that the network we construct use separate channels for the inputs and outputs, the model is equivalent after scaling. Actually the property above allows the tradeoff between $B_{\text{res}}$ and $B_{\text{out}}$. If all the weights in the residual blocks are scaled by an arbitrary positive constant $c$, and the weight in the last layer $\mathbf{W}_{\text{out}}$ is scaled by $c^{-L}$, the model is still equivalent. We only need to scale the all the weights in this block with $|a_{i,k_j,\boldsymbol{s}_j}|^{1/L}$, setting the sign of the weight in the last layer as $\text{sign}(a_{i,k_j,\boldsymbol{s}_j})$, and place $C_{\mathcal{M}}P$ number of these building blocks in this neural network with residual connections. Since this block always output 0 in the first $D+1$ channels, the order and the placement of the building blocks does not change the output. The last fully connected layer can be simply set to

$$\mathbf{W}_{\text{out}} = [0, \dots, 0, 1], b_{\text{out}} = 0.$$

Combining Proposition F.2 and Lemma F.1, the norm of this ResNeXt we construct satisfy

$$\bar{B}_{\text{res}} \leq \sum_{i=1}^{C_{\mathcal{M}}} \sum_{j=1}^{P} a_{i,k_j,\boldsymbol{s}_j}^{2/L} (2^{2k/L} C_{14} dmL + C_{12} DL)$$

$$\leq \sum_{i=1}^{C_{\mathcal{M}}} \sum_{j=1}^{P} (2^k a_{i,k_j,\boldsymbol{s}_j})^{2/L} (C_{14} dmL + C_{12} DL)$$

$$\leq (C_{\mathcal{M}}P)^{1-2/(pL)} \|\{2^k a_{i,k_j,\boldsymbol{s}_j}\}\|_p^{2/L} (C_{14} dmL + C_{12} DL)$$

$$\leq (C_{10} C_{\text{F}})^{2/L} (C_{\mathcal{M}}P)^{1-2/(pL)} (C_{14} dmL + C_{12} DL),$$

$$\bar{B}_{\text{out}} \leq 1.$$

By scaling all the weights in the residual blocks by $\bar{B}_{\text{res}}^{-1/2}$, and scaling the output layer by $\bar{B}_{\text{res}}^{L/2}$, the network that satisfy (12) can be constructed. $\qquad\square$

Notice that the chart selection part does not introduce error by our way of construction, we only need to sum over the error in Section 4.1 and Section 4.1, and notice that for any $\boldsymbol{x}$, for any linear projection $\phi_i$, the number of B-spline basis functions $M_{m,k,\boldsymbol{s}}$ that is nonzero on $\boldsymbol{x}$ is no more than $m^d \log P$, the approximation error of the constructed neural network can be proved.

## C.5 Constructing a convolution neural network to approximate the target function

In this section, we prove that any feedforward neural network can be realized by a convolution neural network with similar size and norm of parameters. The proof is similar to Theorem 5 in [30].

**Lemma C.6.** *For any feedforward neural network with depth $L'$, width $w'$, input dimension $h$ and output dimension $h'$, for any kernel size $K > 1$, there exists a convolution neural network with depth $L = L' + L_0 - 1$, where $L_0 = \lceil \frac{h-1}{K-1} \rceil$ number of channels $w = 4w'$, and the first dimension of the output equals the output of the feedforward neural network for all inputs, and the norm of the convolution neural network is bounded as*

$$\sum_{\ell=1}^{L} \|\mathbf{W}_\ell\|_{\text{F}}^2 \leq 4 \sum_{\ell=1}^{L'} \|\mathbf{W}'_\ell\|_{\text{F}}^2 + 4w' L_0,$$

*where $\mathbf{W}'_1 \in \mathbb{R}^{w' \times h'}; \mathbf{W}'_\ell \in \mathbb{R}^{w' \times w'}, \ell = 2, \dots, L'-1; \mathbf{W}'_{L'} \in \mathbb{R}^{h' \times w'}$ are the weights in the feedforward neural network, and $\mathbf{W}_1 \in \mathbb{R}^{K \times w \times h}, \mathbf{W}_\ell \in \mathbb{R}^{K \times w \times w}, \ell = 2, \dots, L-1; \mathbf{W}_L \in \mathbb{R}^{K \times h \times w}$ are the weights in the convolution neural network.*

*Proof.* We follow the same method as Oono and Suzuki [30] to construct the CNN that is equivalent to the feedforward neural network. By combining Oono and Suzuki [30] lemma 1 and lemma 2, for

any linear transformation, one can construct a convolution neural network with at most $L_0 = \lceil \frac{h-1}{K-1} \rceil$ convolution layers and 4 channels, where $h$ is the dimension of input, which equals $D + 1$ in our case, such that the first dimension in the output equals the linear transformation, and the norm of all the weights is no more than

$$\sum_{\ell=1}^{L_0} \|\mathbf{W}_\ell\|_F^2 \leq 4L_0, \tag{14}$$

where $\mathbf{W}_\ell$ is the weight of the linear transformation. Putting $w$ number of such convolution neural networks in parallel, a convolution neural network with $L_0$ layers and $4w$ channels can be constructed to implement the first layer in the feedforward neural network.

To implement the remaining layers, one choose the convolution kernel $\mathbf{W}_{\ell+L_0-1}[:, i, j] = [0, \ldots, \mathbf{W}'[i, j], \ldots, 0], \forall 1 \leq i, j \leq w$, and pad the remaining parts with 0, such that this convolution layer is equivalent to the linear layer applied on the dimension of channels. Noticing that this conversion does not change the norm of the parameters in each layer. Adding both sides of (14) by the norm of the $2 - L'$-th layer in both models finishes the proof. $\qquad\square$

# D  Proof of the estimation theory

## D.1  Covering number of a neural network block

**Proposition D.1.** *If the input to a ReLU neural network is bounded by $\|\boldsymbol{x}\|_2 \leq B_{\mathrm{in}}$, the covering number of the ReLU neural network defined in Proposition F.6 is bounded by*

$$\mathcal{N}(\mathcal{F}_{NN}, \delta, \|\cdot\|_2) \leq \left( \frac{B_{\mathrm{in}}(B/L)^{L/2} wL}{\delta} \right)^{w^2 L}.$$

*Proof.* Similar to Proposition F.6, we only consider the case $\|W_\ell\|_F \leq \sqrt{B/L}$. For any $1 \leq \ell \leq L$, for any $W_1, \ldots W_{\ell-1}, W_\ell, W'_\ell, W_{\ell+1}, \ldots W_L$ that satisfy the above constraint and $\|W_\ell - W'_\ell\|_F \leq \epsilon$, define $g(\ldots; W_1, \ldots W_L)$ as the neural network with parameters $W_1, \ldots W_L$, we can see

$$\|g(\boldsymbol{x}; W_1, \ldots W_{\ell-1}, W_\ell, W_{\ell+1}, \ldots W_L) - g(\boldsymbol{x}; W_1, \ldots W_{\ell-1}, W_\ell, W_{\ell+1}, \ldots W_L)\|_2$$
$$\leq (B/L)^{(L-\ell)/2} \|W_\ell - W'_\ell\|_2 \|ReLU(W_{\ell-1} \ldots ReLU(W_1(\boldsymbol{x})))\|_2$$
$$\leq (B/L)^{(L-1)/2} B_{\mathrm{in}} \epsilon.$$

Choosing $\epsilon = \frac{\delta}{L(B/L)^{(L-1)/2}}$, the above inequality is no larger than $\delta/L$. Taking the sum over $\ell$, we can see that for any $W_1, W'_1, \ldots, W_L, W'_L$ such that $\|W_\ell - W'_\ell\|_F \leq \epsilon$,

$$\|g(\boldsymbol{x}; W_1, \ldots W_L) - g(\boldsymbol{x}; W'_1, \ldots W'_L))\|_2 \leq \delta.$$

Finally, observe that the covering number of $W_\ell$ is bounded by

$$\mathcal{N}(\{W : \|W\|_F \leq B\}, \epsilon, \|\cdot\|_F) \leq \left( \frac{2Bw}{\epsilon} \right)^{w^2}. \tag{15}$$

Substituting $B$ and $\epsilon$ and taking the product over $\ell$ finishes the proof. $\qquad\square$

**Proposition D.2.** *If the input to a ReLU convolution neural network is bounded by $\|x\|_2 \leq B_{\mathrm{in}}$, the covering number of the ReLU neural network defined in (5) is bounded by*

$$\mathcal{N}(\mathcal{F}_{NN}, \delta, \|\cdot\|_2) \leq \left( \frac{B_{\mathrm{in}}(BK/L)^{L/2} wL}{\delta} \right)^{w^2 KL}.$$

*Proof.* Similar to Proposition D.1, for any $1 \leq \ell \leq L$, for any $W_1, \ldots W_{\ell-1}, W_\ell, W'_\ell, W_{\ell+1}, \ldots W_L$ that satisfy the above constraint and $\|W_\ell - W'_\ell\|_F \leq \epsilon$, define $g(\ldots; W_1, \ldots W_L)$ as the neural network with parameters $W_1, \ldots W_L$, we can see

$$\|g(\boldsymbol{x}; W_1, \ldots W_{\ell-1}, W_\ell, W_{\ell+1}, \ldots W_L) - g(\boldsymbol{x}; W_1, \ldots W_{\ell-1}, W_\ell, W_{\ell+1}, \ldots W_L)\|_2$$
$$\leq K^{L/2}(B/L)^{(L-\ell)/2} \|W_\ell - W'_\ell\|_2 \|ReLU(W_{\ell-1} \ldots ReLU(W_1(\boldsymbol{x})))\|_2$$
$$\leq K^{L/2}(B/L)^{(L-1)/2} B_{\mathrm{in}} \epsilon,$$

where the first inequality comes from Proposition F.10. Choosing $\epsilon = \frac{\delta}{K^{L/2} B_{\text{in}} L (B/L)^{(L-1)/2}}$, the above inequality is no larger than $\delta/L$. Taking this into (15) finishes the proof. $\qquad\square$

## D.2 Proof of Theorem 3.3

Define $\tilde{f} = \arg\min_f \mathbb{E}_{\mathcal{D}}[\mathcal{L}(f)]$. From Theorem 14.20 in Wainwright [36], for any function class $\partial\mathcal{F}$ that is star-shaped around $\tilde{f}$, the empirical risk minimizer $\hat{f} = \arg\min_{f \in \mathcal{F}} \mathcal{L}_n(f)$ satisfy

$$\mathbb{E}_{\mathcal{D}}[\mathcal{L}(\hat{f})] \leq \mathbb{E}_{\mathcal{D}}[\mathcal{L}(\tilde{f})] + 10\delta_n(2 + \delta_n) \tag{16}$$

with probability at least $1 - c_1 \exp(-c_2 n \delta_n^2)$ for any $\delta_n$ that satisfy (20), where $c_1, c_2$ are universal constants.

The function of neural networks is not star-shaped, but can be covered by a star-shaped function class. Specifically, let $\{f - \tilde{f} : f \in \mathcal{F}^{\text{Conv}}\} \subset \{f_1 - f_2 : f_1, f_2 \in \mathcal{F}^{\text{Conv}}\} := \partial\mathcal{F}$.

Any function in $\partial\mathcal{F}$ can be represented using a ResNeXt: one can put two neural networks of the same structure in parallel, adjusting the sign of parameters in one of the neural networks and summing up the result, which increases $M, B_{\text{res}}$ and $B_{\text{out}}$ by a factor of 2. This only increases the log covering number in (11) by a factor of constant (remind that $B_{\text{res}} = O(1)$ by assumption).

Taking the log covering number of the ResNeXt (11), the sufficient condition for the critical radius as in (20) is

$$n^{-1/2} w L^{1/2} B_{\text{res}}^{\frac{1}{2-4/L}} K^{\frac{1-1/L}{1-2/L}} \left(B_{\text{out}}^{1/2} \exp((K B_{\text{res}}/L)^{L/2})\right)^{\frac{1/L}{1-2/L}} \delta_n^{\frac{1-3/L}{1-2/L}} \lesssim \frac{\delta_n^2}{4},$$

$$\delta_n \gtrsim K(w^2 L)^{\frac{1-2/L}{2-2/L}} B_{\text{res}}^{\frac{1}{2-2/L}} \left(B_{\text{out}}^{1/2} \exp((K B_{\text{res}}/L)^{L/2})\right)^{\frac{1/L}{1-1/L}} n^{-\frac{1-2/L}{2-2/L}}, \tag{17}$$

where $\lesssim$ hides the logarithmic term.

Because $\mathcal{L}$ is 1-Lipschitz, we have

$$\mathcal{L}(f) \leq \mathcal{L}(\tilde{f}) + \|f - \tilde{f}\|_\infty.$$

Choosing

$$P = O\left(\left(\frac{K^{-\frac{2}{L-2}} w^{\frac{3L-4}{L-2}} L^{\frac{3L-2}{L-2}}}{n}\right)^{-\frac{1-2/L}{2\alpha/d(1-1/L)+1-2/pL}}\right),$$

and taking Theorem 3.2 and (17) into (16) finishes the proof.

# E Lower bound of error

In this section, we study the minimax lower bound of any estimator for Besov functions on a $d$-dimensional manifold. It suffices to consider the manifold $\mathcal{M}$ as a $d$-dimensional hypersurface. Without the loss of generality, assume that $\frac{\partial\mathcal{L}(y)}{\partial y} \geq 0.5$ for $-\epsilon \leq y \leq \epsilon$. Define the function space

$$\mathcal{F} = \left\{f = \sum_{j_1,\ldots,j_d=1}^{s} \pm\frac{\epsilon}{s^\alpha} \times M^{(m)}((\boldsymbol{x} - \boldsymbol{j})/s)\right\}, \tag{18}$$

where $M^{(m)}$ denotes the Cardinal B-spline basis function that is supported on $(0,1)^d$, $\boldsymbol{j} = [j_1, \ldots, j_d]$. The support of each B-spline basis function splits the space into $s^d$ number of blocks, where the target function in each block has two choices (positive or negative), so the total number of different functions in this function class is $|\mathcal{F}| = 2^{s^d}$. Using Dũng [11, Theorm 2.2], we can see that for any $f \in \mathcal{F}$,

$$\|f\|_{B_{p,q}^\alpha} \leq \frac{\epsilon}{s^\alpha} s^{\alpha-d/p} s^{d/p} = \epsilon.$$

For a fixed $f^* \in \mathcal{F}$, let $\mathcal{D} = \{(\boldsymbol{x}_i, y_i)\}_{i=1}^n$ be a set of noisy observations with $y_i = f^*(\boldsymbol{x}_i) + \epsilon_i, \epsilon_i \sim SubGaussian(0, \sigma^2 I)$. Further assume that $\boldsymbol{x}_i$ are evenly distributed in $(0,1)^d$ such that in all

regions as defined in (18), the number of samples is $n_{\boldsymbol{j}} := O(n/s^d)$. Using Le Cam's inequality, we get that in any region, any estimator $\theta$ satisfy

$$\sup_{f^* \in \mathcal{F}} \mathbb{E}_{\mathcal{D}}[\|\theta(\mathcal{D}) - f^*\|_{\boldsymbol{j}}] \geq \frac{C_m \epsilon}{16 s^\alpha}$$

as long as $(\frac{\epsilon}{\sigma s^\alpha})^2 \lesssim \frac{s^d}{n}$, where $\| \cdot \|_{\boldsymbol{j}} := \frac{1}{n_{\boldsymbol{i}}} \sum_{s(\boldsymbol{x}-\boldsymbol{j})\in[0,1]^d} |f(\boldsymbol{x})|$ denotes the norm defined in the block indexed by $\boldsymbol{i}$, $C_m$ is a constant that depends only on $m$. Choosing $s = O(n^{\frac{1}{2\alpha+d}})$, we get

$$\sup_{f^* \in \mathcal{F}} \mathbb{E}_{\mathcal{D}}[\|\theta(\mathcal{D}) - f^*\|_{\boldsymbol{j}}] \geq n^{-\frac{\alpha}{2\alpha+d}}.$$

Observing $\frac{1}{n} \sum_{i=1}^n L(\hat{(f}(\boldsymbol{x}_i))) \geq 0.5 \sum_{i=1}^n |f(\boldsymbol{x}_i) - f^*(\boldsymbol{x}_i)| \asymp \frac{1}{s^d} \sum_{\boldsymbol{j}\in[s]^d} \|\hat{f} - f^*\|_{\boldsymbol{j}}$ finishes the proof.

# F    Supporting theorem

**Lemma F.1.** *[Lemma 14 in Zhang and Wang [43]] For any $a \in \mathbb{R}^{\bar{M}}$, $0 < p' < p$, it holds that:*

$$\|a\|_{p'}^{p'} \leq \bar{M}^{1-p'/p} \|a\|_p^{p'}.$$

**Proposition F.2** (Proposition 7 in Zhang and Wang [43]). *Let $\alpha - d/p > 1, r > 0$. For any function in Besov space $f^* \in B_{p,q}^\alpha$ and any positive integer $\bar{M}$, there is an $\bar{M}$-sparse approximation using B-spline basis of order $m$ satisfying $0 < \alpha < \min(m, m - 1 + 1/p)$: $\check{f}_{\bar{M}} = \sum_{i=1}^{\bar{M}} a_{k_i,\boldsymbol{s}_i} M_{m,k_i,\boldsymbol{s}_i}$ for any positive integer $\bar{M}$ such that the approximation error is bounded as $\|\check{f}_{\bar{M}} - f^*\|_r \lesssim \bar{M}^{-\alpha/d} \|f^*\|_{B_{p,q}^\alpha}$, and the coefficients satisfy*

$$\|\{2^{k_i} a_{k_i,\boldsymbol{s}_i}\}_{k_i,\boldsymbol{s}_i}\|_p \lesssim \|f^*\|_{B_{p,q}^\alpha}.$$

**Lemma F.3** (Lemma 11 in [43]). *Let $M_{m,k,\boldsymbol{s}}$ be the B-spline of order $m$ with scale $2^{-k}$ in each dimension and position $\boldsymbol{s} \in \mathbb{R}^d$: $M_{m,k,\boldsymbol{s}}(\boldsymbol{x}) := M_m(2^k(\boldsymbol{x} - \boldsymbol{s}))$, $M_m$ is defined in (2). There exists a neural network with $d$-dimensional input and one output, with width $w_{d,m} = O(dm)$ and depth $L \lesssim \log(C_{13}/\epsilon)$ for some constant $C_{13}$ that depends only on $m$ and $d$, approximates the B spline basis function $M_{m,k,\boldsymbol{s}}(\boldsymbol{x}) := M_m(2^k(\boldsymbol{x} - \boldsymbol{s}))$. This neural network, denoted as $\tilde{M}_{m,k,\boldsymbol{s}}(\boldsymbol{x}), \boldsymbol{x} \in \mathbb{R}^d$, satisfy*

- $|\tilde{M}_{m,k,\boldsymbol{s}}(\boldsymbol{x}) - M_{m,k,\boldsymbol{s}}(\boldsymbol{x})| \leq \epsilon$, *if $0 \leq 2^k(x_i - s_i) \leq m + 1, \forall i \in [d]$,*

- $\tilde{M}_{m,k,\boldsymbol{s}}(\boldsymbol{x}) = 0$, *otherwise.*

- *The total square norm of the weights is bounded by $2^{2k/L} C_{14} dmL$ for some universal constant $C_{14}$.*

**Proposition F.4.** *For any feedforward neural network $f$ with width $w$ and depth $L$ with bias, there exists a feedforward neural network $f'$ with width $w' = w + 1$ and depth $L' = L$, such that for any $\boldsymbol{x}$, $f(\boldsymbol{x}) = f'([\boldsymbol{x}^T, 1]^T)$*

*Proof.* Proof by construction: let the weights in the $\ell$-th layer in $f$ be $\mathbf{W}_\ell$, and the bias be $\boldsymbol{b}_\ell$, and choose the weight in the corresponding layer in $f'$ be

$$\mathbf{W}'_\ell = \begin{bmatrix} \tilde{\mathbf{W}}_\ell & \tilde{\boldsymbol{b}}_\ell \\ 0 & 1 \end{bmatrix}, \quad \forall \ell < L; \quad \mathbf{W}'_L = [\tilde{\mathbf{W}}_L \quad \tilde{\boldsymbol{b}}_L].$$

The constructed neural network gives the same output as the original one. $\square$

**Corollary F.5** (Corollary 13.7 and Corollary 14.3 in Wainwright [36]). *Let*

$$\mathcal{G}_n(\delta, \mathcal{F}) = \mathbb{E}_{w_i}\left[\sup_{g\in\mathcal{F}, \|g\|_n\leq\delta} \left|\frac{1}{n}\sum_{i=1}^n w_i g(\boldsymbol{x}_i)\right|\right], \mathcal{R}_n(\delta, \mathcal{F}) = \mathbb{E}_{\epsilon_i}\left[\sup_{g\in\mathcal{F}, \|g\|_n\leq\delta} \left|\frac{1}{n}\sum_{i=1}^n \epsilon_i g(\boldsymbol{x}_i)\right|\right],$$

denotes the local Gaussian complexity and local Rademacher complexity respectively, where $w_i \sim \mathcal{N}(0, 1)$ are the i.i.d. Gaussian random variables, and $\epsilon_i \sim \text{uniform}\{-1, 1\}$ are the Rademacher random variables. Suppose that the function class $\mathcal{F}$ is star-shaped, for any $\sigma > 0$, any $\delta \in (0, \sigma]$ such that

$$\frac{16}{\sqrt{n}} \int_{\delta_n^2/4\sigma}^{\delta_n} \sqrt{\log \mathcal{N}(\mathcal{F}, \mu, \|\cdot\|_\infty)} d\mu \leq \frac{\delta_n^2}{4\sigma}$$

satisfies

$$\mathcal{G}_n(\delta, \mathcal{F}) \leq \frac{\delta^2}{2\sigma}. \tag{19}$$

Furthermore, if $\mathcal{F}$ is uniformly bounded by b, i.e. $\forall f \in \mathcal{F}, \boldsymbol{x}|f(\boldsymbol{x})| \leq b$ any $\delta > 0$ such that

$$\frac{64}{\sqrt{n}} \int_{\delta_n^2/2b4\sigma}^{\delta_n} \sqrt{\log \mathcal{N}(\mathcal{F}, \mu, \|\cdot\|_\infty)} d\mu \leq \frac{\delta_n^2}{b}.$$

satisfies

$$\mathcal{R}_n(\delta, \mathcal{F}) \leq \frac{\delta^2}{b}. \tag{20}$$

**Proposition F.6.** *An $L$-layer ReLU neural network with no bias and bounded norm*

$$\sum_{\ell=1}^{L} \|\mathbf{W}_\ell\|_{\mathrm{F}}^2 \leq B$$

*is Lipschitz continuous with Lipschitz constant $(B/L)^{L/2}$*

*Proof.* Notice that ReLU function is 1-homogeneous, similar to Proposition 4 in [43], for any neural network there exists an equivalent model satisfying $\|\mathbf{W}_\ell\|_{\mathrm{F}} = \|\mathbf{W}_{\ell'}\|_{\mathrm{F}}$ for any $\ell, \ell'$, and its total norm of parameters is no larger than the original model. Because of that, it suffices to consider the neural network satisfying $\|\mathbf{W}_\ell\|_{\mathrm{F}} \leq \sqrt{B/L}$ for all $\ell$. The Lipschitz constant of such linear layer is $\|\mathbf{W}_\ell\|_2 \leq \|\mathbf{W}_\ell\|_{\mathrm{F}} \leq \sqrt{B/L}$, and the Lipschitz constant of ReLU layer is 1. Taking the product over all layers finishes the proof. $\square$

**Proposition F.7.** *An $L$-layer ReLU convolution neural network with convolution kernel size $K$, no bias and bounded norm*

$$\sum_{\ell=1}^{L} \|\mathbf{W}_\ell\|_{\mathrm{F}}^2 \leq B.$$

*is Lipschitz continuous with Lipschitz constant $(KB/L)^{L/2}$*

This proposition can be proved by taking Proposition F.10 into the proof of Proposition F.6.

**Proposition F.8.** *Let $f = f_{\mathrm{post}} \circ (1 + f_{\mathrm{NN}} + f_{\mathrm{other}}) \circ f_{\mathrm{pre}}$ be a ResNeXt, where $1 + f_{\mathrm{NN}} + f_{\mathrm{other}}$ denotes a residual block, $f_{\mathrm{pre}}$ and $f_{\mathrm{post}}$ denotes the part of the neural network before and after this residual block, respectively. $f_{\mathrm{NN}}$ denotes one of the building block in this residual block and $f_{\mathrm{other}}$ denotes the other residual blocks. Assume $f_{\mathrm{pre}}, f_{\mathrm{NN}}, f_{\mathrm{post}}$ are Lipschitz continuous with Lipschitz constant $L_{\mathrm{pre}}, L_{\mathrm{NN}}, L_{\mathrm{post}}$ respectively. Let the input be $x$, if the residual block is removed, the perturbation to the output is no more than $L_{\mathrm{pre}} L_{\mathrm{NN}} L_{\mathrm{post}} \|\boldsymbol{x}\|$*

*Proof.*

$$|f_{\mathrm{post}} \circ (1 + f_{\mathrm{NN}} + f_{\mathrm{other}}) \circ f_{\mathrm{pre}}(\boldsymbol{x}) - f_{\mathrm{post}} \circ (1 + f_{\mathrm{other}}) \circ f_{\mathrm{pre}}(\boldsymbol{x})|$$
$$\leq L_{\mathrm{post}} |(1 + f_{\mathrm{NN}} + f_{\mathrm{other}}) \circ f_{\mathrm{pre}}(\boldsymbol{x}) - (1 + f_{\mathrm{other}}) \circ f_{\mathrm{pre}}(\boldsymbol{x})|$$
$$= L_{\mathrm{post}} |f_{\mathrm{NN}} \circ f_{\mathrm{pre}}(\boldsymbol{x})|$$
$$\leq L_{\mathrm{pre}} L_{\mathrm{NN}} L_{\mathrm{post}} \|\boldsymbol{x}\|.$$

$\square$

**Proposition F.9.** *The neural network defined in Lemma 4.1 with arbitrary number of blocks has Lipschitz constant $\exp((KB_{\mathrm{res}}/L)^{L/2})$, where $K = 1$ when the feedforward neural network is the building blocks and $K$ is the size of the convolution kernel when the convolution neural network is the building blocks.*

*Proof.* Note that the $m$-th block in the neural network defined in Lemma 4.1 can be represented as $y = f_m(\boldsymbol{x}; \omega_m) + \boldsymbol{x}$, where $f_m$ is an $L$-layer feedforward neural network with no bias. By Proposition F.6 and Proposition F.7, such block is Lipschitz continuous with Lipschitz constant $1 + (KB_m/L)^{L/2}$, where the weight parameters of the $m$-th block satisfy that $\sum_{\ell=1}^{L} \|W_\ell^{(m)}\|_{\mathrm{F}}^2 \leq B_m$ and $\sum_{m=1}^{M} B_m \leq B_{\mathrm{res}}$.

Since the neural network defined in Lemma 4.1 is a composition of $M$ blocks, it is Lipschitz with Lipschitz constant $L_{\mathrm{res}}$. We have

$$L_{\mathrm{res}} \leq \prod_{m=1}^{M} \left( 1 + \left( \frac{KB_m}{L} \right)^{L/2} \right) \leq \exp \left( \sum_{m=1}^{M} \left( \frac{KB_m}{L} \right)^{L/2} \right),$$

where we use the inequality $1 + z \leq \exp(x)$ for any $x \in \mathrm{R}$. Furthermore, notice that $\sum_{m=1}^{M} (KB_m/L)^{L/2}$ is convex with respect to $(B_1, B_2, \ldots, B_M)$ when $L > 2$. Since $\sum_{m=1}^{M} B_m \leq B_{\mathrm{res}}$ and $B_m \geq 0$, then we have $\sum_{m=1}^{M} (KB_m/L)^{L/2} \leq (KB_{\mathrm{res}}/L)^{L/2}$ by convexity. Therefore, we obtain that $L_{\mathrm{res}} \leq \exp((KB_{\mathrm{res}}/L)^{L/2})$. $\square$

**Proposition F.10.** *For any $\boldsymbol{x} \in \mathbb{R}^d, \boldsymbol{w} \in \mathbb{R}^K, K \leq d, \|\mathrm{Conv}(\boldsymbol{x}, \boldsymbol{w})\|_2 \leq \sqrt{K} \|\boldsymbol{x}\|_2 \|\boldsymbol{w}\|_2.$*

*Proof.* For simplicity, denote $x_i = 0$ for $i \leq 0$ or $i > d$.

$$\begin{aligned}
\|\mathrm{Conv}(\boldsymbol{x}, \boldsymbol{w})\|_2^2 &= \sum_{i=1}^{d} \langle \boldsymbol{x}[i - \tfrac{K-1}{2} : i + \tfrac{K-1}{2}], \boldsymbol{w} \rangle^2 \\
&\leq \sum_{i=1}^{d} \|\boldsymbol{x}[i - \tfrac{K-1}{2} : i + \tfrac{K-1}{2}]\|_2^2 \|\boldsymbol{w}\|_2^2 \\
&\leq K \|\boldsymbol{x}\|_2^2 \|\boldsymbol{w}\|_2^2,
\end{aligned}$$

where the second line comes from Cauchy-Schwarz inequality, the third line comes by expanding $\|\boldsymbol{x}[i - \tfrac{K-1}{2} : i + \tfrac{K-1}{2}]\|_2^2$ by definition and observing that each element in $\boldsymbol{x}$ appears at most $K$ times. $\square$

