# OpenReview forum: "Nonparametric Classification on Low Dimensional Manifolds using Overparameterized Convolutional Residual Networks"
_NeurIPS.cc/2024/Conference — NeurIPS 2024 poster_

### Official Review · Reviewer_giSF · 2024-07-02

**Soundness:** 4
**Presentation:** 4
**Contribution:** 3
**Rating:** 8
**Confidence:** 5

**Summary:**

The authors study the performance of ConvResNets trained with weight decay from the perspective of nonparametric classification. Specifically, the authors consider a smooth target function supported on a low-dimensional manifold, then prove that ConvResNets can adapt to the function smoothness and low-dimensional structures and efficiently learn the function without suffering from the curse of
dimensionality.

**Strengths:**

This is an excellent, novel and strong work. The paper provides theoretical guarantees for the ConvResNets in the setting of nonparametric classification. The minimax optimal rate under ConvResNets has been an open problem which was successfully resolved by the authors in the conditional class probability.

**Weaknesses:**

Literature review is not sufficient. Minimax optimal nonparametric classification using DNN has been recently studied by other authors, though the current paper consider another novel neural network, the ConvResNets, which is different from the traditional DNN.

**Questions:**

Nonparametric classification can be done in different frameworks such as (1) smooth decision boundary, (2) smooth conditional class probability or (3) smooth margin. The minimax optimal rates in the above settings are different. For instance, some authors find that DNN classification in (1) and (2) yield different minimax rates.  In particular, the minimax optimal rate under (1) is an open problem (see Kim's work) which was recently addressed by Hu et al. The minimax rate under (2) has been recently established for ultrahigh-dimensional case for DNN classification by Wang's work. The minimax rates under (1) and (2) may have different expressions.

I guess that this paper considers the setting of smooth conditional class probability, and the manifold dimension D is fixed. It would be great to comment on its extension to ultrahigh-dimensional case as in Wang's work.

---

> ### Author Rebuttal · Authors · 2024-08-07
>
> Thank you for taking the time to review the paper and for your positive feedback. I'm grateful for your engagement and would like to address the question you've raised with the following response.
>
>
> **Weaknesses: Literature review is not sufficient. Minimax optimal nonparametric classification using DNN has been recently studied by other authors, though the current paper considers another novel neural network, the ConvResNets, which is different from the traditional DNN.**
>
> Due to space limits, we are only able to briefly mention the existing literature on FNNs/DNNs in Line 23-26. We will add more details of literature in the next version.
>
>
> **Q: Nonparametric classification can be done in different frameworks such as (1) smooth decision boundary, (2) smooth conditional class probability or (3) smooth margin… The minimax rate under (2) has been recently established for ultrahigh-dimensional case for DNN classification by Wang's work… I guess that this paper considers the setting of smooth conditional class probability, and the manifold dimension D is fixed. It would be great to comment on its extension to ultrahigh-dimensional case as in Wang's work.**
>
> A: Thank you for the reference [1] and your insightful question! You are correct that our paper focuses on the setting of smooth conditional class probability (2), with a fixed manifold dimension. We will add further discussions about [1] in our next version.
>
> The ultrahigh-dimensional case would be an interesting extension of our work! Regarding the specific Wang’s paper you mentioned, we’re currently reviewing [2] and [3] as they appear relevant. While neither paper explicitly defines "ultrahigh-dimension," [3] characterizes it as a scenario where the dimension $D$ grows at a non-polynomial rate relative to the sample size $n$, though it does not establish a minimax convergence rate. Our statistical rate avoids exponential dependence in $D$. However, to achieve a convergence rate that logarithmically depends on $D$, additional assumptions on input features are required, such as having only a small fraction of the variables active for the outputs.
>
> We would greatly appreciate any further clarification on the reference and the definition of ultrahigh-dimension. We are eager to delve deeper into this topic and discuss it further.
>
> Reference:
>
> [1] Kim, Yongdai, Ilsang Ohn, and Dongha Kim. "Fast convergence rates of deep neural networks for classification." Neural Networks 138 (2021): 179-197.
>
> [2] Wang H, Jin X, Wang J, Hao H. Nonparametric Estimation for High-Dimensional Space Models Based on a Deep Neural Network. Mathematics. 2023; 11(18):3899.
>
> [3] Li, K., Wang, F., Yang, L., & Liu, R. (2023). Deep feature screening: Feature selection for ultra high-dimensional data via deep neural networks. Neurocomputing, 538, 126186.

---

> > ### Comment · Reviewer_giSF · 2024-08-11
> > **Thanks for responses!**
> >
> > My concerns were well addressed unless some other references in DNN classification are still missing. For instance, I notice the following works are relevant:
> >
> > Minimax optimal deep neural network classifiers under smooth decision boundary
> >
> > Sharp rate of convergence for deep neural network classifiers under the teacher-student setting
> >
> > Deep neural network classifier for multidimensional functional data
> >
> > Minimax optimal high‐dimensional classification using deep neural networks

---

> > > ### Author Response · Authors · 2024-08-12
> > >
> > > Thank you for the references! We will cite these works and add further discussions in our next version.

---

### Official Review · Reviewer_xShA · 2024-07-13

**Soundness:** 2
**Presentation:** 3
**Contribution:** 2
**Rating:** 4
**Confidence:** 4

**Summary:**

The paper studies the performance of an overparametrized convolutional residual network architecture on a nonparametric classification problem trained with weight decay. This model, known as ConvResNeXt, involves N residual blocks, where each residual block has a parallel architecture of M building blocks, and each building block with L layers. As such it generalizes alternative ConvResNets in the literature and includes them as a special case, e.g., ConvResNet [He et al 2016] when M=1 and aggregated Resnet [Xie et at 2017] when N=1. Specifically, the analysis results consider infinitely many building blocks and shows that weight decay training implicitly enforces sparsity of these blocks considering that the optimal classifier is supported on a low dimensional smooth manifold. As such, they show that this model can efficiently learn the assumed function class without suffering from the  curse of dimensionality.

**Strengths:**

1)	The paper shows that for learning Besov functions trained with weight decay out of n samples, the rate of convergence depends only on the intrinsic low dimension enabling the curse of dimensionality.
2)	They also provide a tighter bound for the weight-decayed ConvResNeXt in computing the critical radius of local Gaussian complexity.

**Weaknesses:**

1) The actual benefit of having parallel paths in the proposed ConvResNeXt model was not addressed fully. Indeed, the results stating that only the product of number of parallel paths M and depth N determines the convergence rate, and one may not need parallel blocks when residual connections are present contradict with some of the earlier findings in the literature. For example, in [Xie et at 2017]. In the mentioned prior work, several experimental results focus on the posive effect of having higher cardinality (more multiple parallel paths) than increased depth and width.
2) If there is no additional benefit of having additional parallel paths, and an equivalent performance can be achieved by increasing the depth only, this will limit the capability of the proposed ConvResNeXt over the ConvResNets unline the claimed benefits of the former.

**Questions:**

1)	This new architecture introduce significantly more complex nested function form, but similar performance in both approximation and estimation errors can be achieved by rather simpler one. The provided bounds on the two errors do not suggest any practical benefit of ConvResNeXts over ConvResNets. If so, what would be the motivation to generalize for the former?
2)	In Sec 3.2, it was stated that the number of building blocks M does not influence the estimation error as long as it is large enough. This must be contrasted with the earlier findings about the cardinality of the parallel paths, and any difference must be clearly stated.
3)	In Sec 3.2, the tradeoff between width and depth was mentioned to be less important in ResNext. In particular, it was stated that the lower bound on the error does not depend on the arrangements of the blocks M and N as long as their product remains same and large enough. One should further address the reason behind this and compare it with the literature again. Moreover, if that’s so, does this suggest that an equivalent model that can achieve equivalent performance without parallel paths?

**Limitations:**

No, authors did not adequately address the limitations of their work, and no potential negative societal impact of their work is identified.

---

> ### Author Rebuttal · Authors · 2024-08-07
>
> Thank you for taking the time to review the paper. I would like to address the concerns you raised in your review.
>
> **Weakness: The actual benefit of having parallel paths was not addressed fully… In Xie et at 2017, several experimental results focus on the positive effect of having higher cardinality (more multiple parallel paths)... this will limit the capability of the proposed ConvResNeXt over the ConvResNets unline the claimed benefits of the former.**
>
> We would like to emphasize that our paper focuses on the representation and generalization abilities of overparameterized ConvResNeXts, not their optimization. We demonstrate that **the statistical rate and representation power are equivalent** whether increasing the number of blocks (M) or paths (N), as long as MN exceeds a certain threshold (Theorem 3.3). This novel finding from approximation and generalization theory perspectives explains the practical success of overparameterized ConvResNeXts.
>
> Practical performance involves factors beyond our theoretical scope, including optimization aspects and techniques like layer normalization which are barely considered by theorists. Xie et al. 2017 used FLOPs as a complexity measure, while we consider $MN$ values. Notably, their results showing comparable performance for ResNeXts with similar MN values still align with our findings.
>
> We acknowledge the potential practical benefits of parallel paths. Our theories suggest this advantage does **not** stem from improved representation or generalization capabilities, but likely from optimization aspects. ResNeXts with more parallel paths may be easier to learn with certain algorithms, but this warrants further investigation. Our approach, decoupling learning and optimization, follows the tradition of Vapnik, Bartlett, and other learning theorists. It allows for a more fine-grained learning theory, complementing optimization-focused studies. We develop generalization bounds for regularized ERM, which address overfitting and potentially unifying optimization and statistical theories in future research.
>
> **Q1: This new architecture introduces significantly more complex nested function form... The provided bounds on the two errors do not suggest any practical benefit of ConvResNeXts over ConvResNets. If so, what would be the motivation to generalize for the former?**
>
> A1: Our paper not only generalizes ConvResNets to ConvResNeXts, but more importantly, we address the ***overparameterization*** regime where parameters can significantly outnumber samples. This contrasts with previous work like Liu et al., which imposes unavoidable cardinality constraints on block numbers, limiting their ability to explain the success of overparameterized networks in practice.
>
> Analyzing overparameterized ConvResNeXts presents unique theoretical challenges, particularly in ***bounding metric entropy***. We tackle this using an advanced method leveraging Dudley's chaining of metric entropy (via critical radius / local Gaussian/Rademacher complexity, Bartlett et al., 2005; see Corollary F.5, Line 1100, Page 21).
>
> Understanding complex networks is inherently difficult. Our theory takes significant steps towards explaining real-world applications, bridging the gap between theoretical analysis and practical success of ***overparameterized*** models. While our bounds don't suggest immediate practical benefits of ConvResNeXts over ConvResNets, they provide crucial insights into the behavior of complex, highly parameterized architectures that dominate modern deep learning practice.
>
> **Q2: In Sec 3.2, it was stated that the number of building blocks M does not influence the estimation error as long as it is large enough. This must be contrasted with the earlier findings about the cardinality of the parallel paths, and any difference must be clearly stated.**
>
> A2: Our findings differ from previous work like Liu et al. 2021, where network complexity and optimal estimation error required careful bounding of cardinality. In our paper, we don't constrain the number of blocks or paths in ConvResNeXts, thanks to ***weight decay*** training.
>
> Weight decay effectively bounds the weights' norms (line 254). As shown in Lemma 4.1, this induces weak sparsity in ConvResNeXt architectures, where many blocks have negligible influence on the network output post-training. Consequently, only a finite number of blocks with significant weight norms contribute meaningfully, allowing us to bound the complexity of overparameterized ConvResNeXts.
>
> This approach enables us to achieve optimal estimation error **without** explicit constraints on network structure. We'll elaborate on the benefits of weight decay in our revised version.
>
> **Q3: In Sec 3.2, the tradeoff between width and depth was mentioned to be less important in ResNext... One should further address the reason behind this and compare it with the literature again. Moreover, if that’s so, does this suggest that an equivalent model that can achieve equivalent performance without parallel paths?**
>
> A3: Our analysis shows that the arrangements of blocks ($M$) and paths ($N$) don't affect the representation and generalization ability of overparameterized ConvResNeXts, as long as their product ($MN$) remains sufficiently large. Theoretically, this is because:
>
> 1. ConvResNeXts with the same MN have equivalent representation power, regardless of M and N arrangements.
>
> 2. These arrangements don't significantly influence network complexity or generalization capability.
>
> Our findings suggest that an equivalent model without parallel paths could achieve similar ***representation*** and ***generalization*** results. However, practical performance also depends on optimization aspects, which our paper doesn't address directly.
> Our work provides valuable insights into approximation and generalization that can't be obtained solely from an optimization perspective.

---

> > ### Author Response · Authors · 2024-08-11
> >
> > Dear Reviewer,
> >
> > Thank you again for the detailed review. As there are only two days left in the discussion period, we wanted to ask if you have any further comments or questions for us to consider.
> >
> > If our rebuttal discussing our result's novelty and contributions addressed your concerns, we would really appreciate it if you would consider raising your score.
> >
> > Best,
> >
> > Submission7950 Authors

---

> > ### Comment · Reviewer_xShA · 2024-08-13
> >
> > I thank the authors for detailed response in the rebuttal. After reading all the comments and the paper again, I still have the same concerns regarding the novelty and the contribution of this work.
> > The authors claim that they focus on the representation and generalization abilities of overparameterized ConvResNeXts (which is not sufficiently motivated over the ConvResNets), and not their optimization. However, they state that the potential practical benefits of parallel paths does not stem from improved representation or generalization capabilities, but likely from optimization aspects which they do not cover. I believe this induces a contradiction to their initial motivation if not misinterpretation. Moreover, their theoretical results also reflect the same argument that as long as the product of number of paths and the depth remains the same, the architecture can be reduced down to ConvResNets (with residual connections but no parallel paths). Although I find the discussion and the results interesting, I think it needs further clarification on the motivation and the actual benefit of the proposed model over the existing one by exploring the optimization aspects of their architecture with parallel paths. As such, I would keep my initial rating.

---

> > > ### Author Response · Authors · 2024-08-13
> > >
> > > Thank you for your reply and letting us know your concerns! We understand your main arguments and would like to explain our motivations here. As noted in Section 1.1 and our answer to Q1, our biggest motivation is to provide new theoretical understanding of **overparameterized** networks. Specifically, we establish representation theory and statistical rate for the **overparameterized** ResNeXts (or ResNets as a special case) when learning Besov functions on low-dimensional manifolds, rather than merely generalizing ResNets to ResNeXts. In comparison, existing results such as Liu et. al. 2021 can only work for ResNets with exact-parameterization, and fails in overparameterized regimes for either ResNeXts or ResNets.
> > >
> > > Moreover, we would like to clarify that we are **not** motivated to either propose new architectures or demonstrate the marginal benefits of parallel paths in ResNeXts. Instead, we focus on “overparameterization”, and attain the equivalent representation and generalization power of ResNeXts with the same $MN$ as a **side product**. Therefore, our results do not ***“induce a contradiction to initial motivation”***. To demonstrate this equivalence observed in ResNeXts, we ran some CIFAR10 experiments with a number of $M,N$ combinations and got the following results:
> > >
> > > | # of Blocks $M$ | # of paths $N$ | Width | Epochs | CIFAR10 Accuracy |
> > > |--------------------|-------------------|--------|----------|-----------------------|
> > > | 8           | 3 | 64 |   450 | 96.38|
> > > | 4 | 6 | 64 | 320 | 96.37|
> > > | 3 | 8 | 64 | 300 | 96.35 (Table 7, Xie et. al. 2017)|
> > >
> > > Our experiments show that ResNeXts with more blocks, if allowed more training efforts, can achieve comparable performance. This also evidentiates that ResNeXts with less parallel paths, though likely harder to be optimized, can possess the same representation and generalization power.
> > >
> > > We hope the reviewer can value our contributions and see the practical implications of our theoretical findings.

---

### Official Review · Reviewer_QKy6 · 2024-07-13

**Soundness:** 3
**Presentation:** 3
**Contribution:** 3
**Rating:** 7
**Confidence:** 3

**Summary:**

This paper builds the deep learning theory for studying convolutional residual neural networks with data lying on an embedded lower-dimensional manifold. Theoretical results on both approximation error and estimation error are provided, and interesting implications from the theory are discussed.

**Strengths:**

1. The paper is well-written and I have enjoyed reading it.
2. The theory is well-developed and clear.
3. Good contributions are made to the deep learning theory.

**Weaknesses:**

1. This paper avoids the curse of dimensionality through assuming that the data lies in a low-dimensional manifold embedded in the ambient space. While real world data typically exhibit manifold structure, they commonly don't lie precisely on a manifold, but instead are surrounding a manifold (i.e. + noise). The theory of this paper highly depends on the exact manifold assumption since it needs the associated partition of unity to split the problem into local charts.
2. Among the remarks/implications made on page 7, it is said that overparameterization is fine. However, to my understanding, the problem of overparameterization is not about an overparameterized model not being expressive enough (which this paper addresses), but really about the difficulty in finding a good enough local minimizer for it during optimization. To this regard, Theorem 3.3 based on the global minimizer doesn't really answer it.

**Questions:**

1. The remarks on tradeoff between width and depth are very interesting. Have you tried this in numerical experiments to see if this theoretical driven intuition actually appears in experiments?
2. Other than the analysis being more general, what are the theoretical benefits of adding the identity map (i.e. the residual network) and multiple CNN blocks together compared to just using a CNN block?

**Limitations:**

See weaknesses and limitations.

---

> ### Author Rebuttal · Authors · 2024-08-07
>
> Thank you for taking the time to review the paper and for your positive feedback. I'm grateful for your engagement and would like to address the question you've raised with the following response.
>
> **Weakness 1: The theory of this paper highly depends on the exact manifold assumption since it needs the associated partition of unity to split the problem into local charts.**
>
> While our paper focuses on the exact manifold setting for clarity, our data assumptions can be relaxed in two significant ways:
>
> 1. Our findings extend to settings where data concentrate on the manifold with orthogonal noise, as suggested by [1].
>
> 2. Further, our results apply to data not necessarily near a manifold, but concentrated around a subset of $\mathrm{R}^D$ with effective Minkowski dimension [2]. In this case, the statistical rate depends only on this effective dimension.
>
> These extensions demonstrate the broader applicability of our theoretical framework beyond the exact manifold assumption, encompassing more general data structures found in practical scenarios.
>
> Reference:
>
> [1] Cloninger, Alexander, and Timo Klock. "A deep network construction that adapts to intrinsic dimensionality beyond the domain." Neural Networks 141 (2021): 404-419.
>
> [2] Zhang, Z., Chen, M., Wang, M., Liao, W., & Zhao, T. (2023, July). Effective minkowski dimension of deep nonparametric regression: function approximation and statistical theories. In International Conference on Machine Learning (pp. 40911-40931). PMLR.
>
>
> **Weakness 2: The problem of overparameterization is not about an overparameterized model not being expressive enough (which this paper addresses), but really about the difficulty in finding a good enough local minimizer for it during optimization. To this regard, Theorem 3.3 based on the global minimizer doesn't really answer it.**
>
> Our paper focuses on **approximation** and **generalization** aspects of overparameterized ConvResNeXts, rather than optimization. We demonstrate that overparameterized ConvResNeXts can adapt to Besov functions on low-dimensional manifolds, achieving near-optimal convergence rates.
>
> While optimization is crucial, existing research [1-4] on this aspect is limited to **simpler networks** or specific function classes. Considering optimization for deep networks often leads to NTK analysis, which suffers from the curse of dimensionality as mentioned in Line 260-261.
>
> Our approach, decoupling learning and optimization, follows the tradition of Vapnik, Bartlett, and other learning theorists. It allows for a more fine-grained learning theory, complementing optimization-focused studies. We develop generalization bounds for regularized ERM, explaining how these mitigate **overfitting**.
>
> We acknowledge that addressing all aspects (approximation, generalization, optimization) for complex networks adapting to Besov functions on low-dimensional structures remains an open challenge. Even so, among all existing works that focus on approximation and generalization, our work is the most closely aligned with real-world practice, considering weight-decay training, overparameterization setting and practical network architectures.
>
> Reference:
>
> [1] Nichani, Eshaan, Alex Damian, and Jason D. Lee. "Provable guarantees for nonlinear feature learning in three-layer neural networks." Advances in Neural Information Processing Systems 36 (2024).
>
> [2] Wang, Zihao, Eshaan Nichani, and Jason D. Lee. "Learning Hierarchical Polynomials with Three-Layer Neural Networks." The Twelfth International Conference on Learning Representations. 2023.
>
> [3] Allen-Zhu, Zeyuan, and Yuanzhi Li. "What can resnet learn efficiently, going beyond kernels?." Advances in Neural Information Processing Systems 32 (2019).
>
> [4] Allen-Zhu, Zeyuan, and Yuanzhi Li. "Backward feature correction: How deep learning performs deep (hierarchical) learning." The Thirty Sixth Annual Conference on Learning Theory. PMLR, 2023.
>
>
> **Q1: The remarks on tradeoff between width and depth are very interesting. Have you tried this in numerical experiments to see if this theoretical driven intuition actually appears in experiments?**
>
> A1: Thank you for this insightful question. Our paper focuses on theoretical analysis, demonstrating that representation and generalization abilities are equivalent for increasing either depth ($M$) or width ($N$), given a sufficiently large product $MN$.
>
> Our findings align with experimental observations in Xie et al. 2017 (Table 3,4), where ResNeXts with similar MN values achieve comparable performance. While they noted slightly better performance for networks with more paths, our theory suggests this advantage doesn't stem from improved representation or generalization capabilities, but may instead be related to optimization aspects.
>
>
> **Q2: What are the theoretical benefits of adding the identity map (i.e. the residual network) and multiple CNN blocks together compared to just using a CNN block?**
>
> A2: Identity maps address vanishing gradients, enabling deeper networks and easier optimization (Xie et. al. 2017). Multiple CNN blocks allow learning hierarchical features at different scales. While we don't prove their theoretical benefits directly, these components are crucial in state-of-the-art architectures (He et al. 2016, Xie et al. 2017). Our work contributes by establishing a close-to-optimal convergence rate for a setting that closely mimics real applications: practical architectures with overparameterization, training with weight decay, and data exhibiting low-dimensional structures, bridging theory and practical success of these complex structures.

---

> > ### Author Response · Authors · 2024-08-11
> >
> > Dear Reviewer,
> >
> > Thank you again for the detailed review. As there are only two days left in the discussion period, we wanted to ask if you have any further comments or questions for us to consider.
> >
> > If our rebuttal discussing our result's generality and contributions addressed your concerns, we would really appreciate it if you would consider raising your score.
> >
> > Best,
> >
> > Submission7950 Authors

---

> > > ### Comment · Reviewer_QKy6 · 2024-08-12
> > >
> > > Thank you to the authors for the detailed reply. Most of my concerns and questions have been addressed, and I will increase my rating accordingly.

---

### Official Review · Reviewer_UzhH · 2024-07-14

**Soundness:** 3
**Presentation:** 3
**Contribution:** 3
**Rating:** 5
**Confidence:** 3

**Summary:**

The paper provide theoretical analysis for the good prediction performance of convolutional residual neural networks, even overparameterized.

**Strengths:**

1. The theory they developed with overparamterized ConvResNeXts trained with weight decay is novel.
2. their theory does not suffer from the curse of dimensionality.

**Weaknesses:**

In line 71, 72, the overparameterization is defined as " the number of blocks is larger than the order of the sample size n". Where is this definition comes from?

**Questions:**

1. Can you generalize from  the empirical logistic risk to other risks? How will the choice of loss change the bound?
2. Can you explain in details about the extra effort (underlying theoretical techinique) you made in this paper to differentiate from  Liu et al. 2021 (Besov function approximation and binary classification on low-dimensional manifolds using convolutional residual networks)? How do you generalize it to over-parameterized setting and ConvResNeXt?

**Limitations:**

The analysis is limited to convolutional networks. It's interensting to see if it can be further generalized to other architectures like transformers.

---

> ### Author Rebuttal · Authors · 2024-08-07
>
> Thank you for taking the time to review the paper. I would like to address the questions you've raised with the following responses.
>
> **Weakness: In line 71, 72, the overparameterization is defined as "the number of blocks is larger than the order of the sample size n". Where is this definition from?**
>
> Thank you for this question. The definition in lines 71-72 is specific to our analysis, not drawn from an existing source. We define overparameterization as 'the number of blocks larger than the order of the sample size' because:
>
> 1. It aligns with the general concept of overparameterization in deep learning, where parameters significantly exceed training samples. This concept is central to recent groundbreaking work [1, 2].
>
> 2. In our ConvResNeXt setting, each block has nearly constant parameters. Thus, when blocks exceed sample size order, total parameters do too.
>
> 3. This definition enables our theoretical analysis of highly expressive models relative to available data.
>
> While not a standard definition, it provides a clear, quantifiable criterion for our work on ConvResNeXts. We'll clarify this in our revised manuscript.
>
> Reference:
>
> [1] Zhang et al. (2017). Understanding deep learning requires rethinking generalization. ICLR.
>
> [2] Belkin et al. (2019). Reconciling modern machine learning practice and the bias-variance trade-off. PNAS.
>
> **Q1: Can you generalize from the empirical logistic risk to other risks? How will the choice of loss change the bound?**
>
> A1: Yes, our findings generalize to other risk functions that exhibit Lipschitz continuity in a bounded domain with respect to model outputs.  While our paper predominantly uses the logistic loss due to its popularity in classification tasks, our results can be extended to a broader range of loss functions. The statistical rate remains the same, differing only by a constant factor that depends on the Lipschitz constant of the chosen loss.
>
> **Q2: Can you explain in details about the extra effort (underlying theoretical technique) you made in this paper to differentiate from Liu et al. 2021? How do you generalize it to over-parameterized settings and ConvResNeXt?**
>
> A2: Our paper extends beyond Liu et al. 2021 in several key ways. We study overparameterized settings and the more complex ConvResNeXt architecture, allowing for flexible network block design. Unlike traditional approaches as adopted by Liu et. al. 2021, we **avoid cardinality constraints** on block numbers through weight decay training, enabling highly parameterized networks. Our analysis of weight decay's effects (line 254 and Lemma 4.1) shows it induces weak sparsity, where only a finite number of blocks significantly contribute. This allows us to bound complexity in overparameterized ConvResNeXts without explicit structural constraints.
>
> A major theoretical challenge we address is **bounding metric entropy** in complex, overparameterized network settings. We employ an advanced method using Dudley's chaining of metric entropy via critical radius / local Gaussian/Rademacher complexity (Bartlett et al., 2005; see our Corollary F.5, Line 1100, Page 21). These innovations enable us to achieve optimal estimation error for overparameterized ConvResNeXts, extending beyond traditional neural network analysis techniques.

---

> > ### Author Response · Authors · 2024-08-11
> >
> > Dear Reviewer,
> >
> > Thank you again for the detailed review. As there are only two days left in the discussion period, we wanted to ask if you have any further comments or questions for us to consider.
> >
> > If our rebuttal discussing our result's novelty and generality addressed your concerns, we would really appreciate it if you would consider raising your score.
> >
> > Best,
> >
> > Submission7950 Authors

---

### Decision · Program_Chairs · 2024-09-25

**Decision:**

Accept (poster)

**Comment:**

The paper investigates the performance of an overparameterized convolutional residual network architecture (ConvResNeXt) on a nonparametric classification problem trained with weight decay. This model generalizes other ConvResNets, including ConvResNet [He et al., 2016] and aggregated ResNet [Xie et al., 2017].

ConvResNeXt consists of N residual blocks, each with a parallel structure of M building blocks, and each building block containing L layers. The analysis focuses on the scenario with infinitely many building blocks.

Authors show that weight decay promotes sparsity in these blocks, leveraging the fact that the optimal classifier resides on a low-dimensional smooth manifold. This allows the model to learn the target function class efficiently, avoiding the curse of dimensionality.

All reviewers agreed on the merits of the paper.